# Rare variant analyses across multiethnic cohorts identify novel genes for refractive error

Anthony M. Musolf[1,81], Annechien E. G. Haarman [2,3,81], Robert N. Luben[4,5], Jue-Sheng Ong [6], Karina Patasova[7], Rolando Hernandez Trapero[8], Joseph Marsh [8], Ishika Jain[1], Riya Jain[1], Paul Zhiping Wang[9], Deyana D. Lewis[1], Milly S. Tedja [2], Adriana I. Iglesias [2], Hengtong Li[10], Cameron S. Cowan[11], Consortium for Refractive Error and Myopia (CREAM)*, Ginevra Biino[12], Alison P. Klein [13], Priya Duggal [14], David A. Mackey[15], Caroline Hayward [8], Toomas Haller [16], Andres Metspalu [16], Juho Wedenoja [17,18], Olavi Pärssinen[19,20], Ching-Yu Cheng [21,22], Seang-Mei Saw[23,24], Dwight Stambolian[25], Pirro G. Hysi [7], Anthony P. Khawaja [4,5], Veronique Vitart [8], Christopher J. Hammond [7], Cornelia M. van Duijn [26,82], Virginie J. M. Verhoeven [2,3,27,82✉], Caroline C. W. Klaver [2,3,11,28,82✉] & Joan E. Bailey-Wilson [1,82✉]

Refractive error, measured here as mean spherical equivalent (SER), is a complex eye condition caused by both genetic and environmental factors. Individuals with strong positive or negative values of SER require spectacles or other approaches for vision correction. Common genetic risk factors have been identified by genome-wide association studies (GWAS), but a great part of the refractive error heritability is still missing. Some of this heritability may be explained by rare variants (minor allele frequency [MAF] ≤ 0.01.). We performed multiple gene-based association tests of mean Spherical Equivalent with rare variants in exome array data from the Consortium for Refractive Error and Myopia (CREAM). The dataset consisted of over 27,000 total subjects from five cohorts of Indo-European and Eastern Asian ethnicity. We identified 129 unique genes associated with refractive error, many of which were replicated in multiple cohorts. Our best novel candidates included the retina expressed *PDCD6IP*, the circadian rhythm gene *PER3*, and *P4HTM*, which affects eye morphology. Future work will include functional studies and validation. Identification of genes contributing to refractive error and future understanding of their function may lead to better treatment and prevention of refractive errors, which themselves are important risk factors for various blinding conditions.

A full list of author affiliations appears at the end of the paper.

Refractive error has become a major worldwide health concern, with the prevalence of the disease, particularly myopia (nearsightedness), becoming more frequent in both the United States[1] and Europe[2] and reaching epidemic proportions in parts of East Asia[3,4]. Refractive error is caused when the optics of the eye fail to project the focal point of light on the retina, causing a blurred image. Myopia is the refractive error mostly resulting from eye elongation, which can lead to serious ocular complications like myopic macular degeneration, glaucoma and retinal detachment[5–8], and is the second most common cause of blindness[9–11].

Refractive error is a highly complex trait that is known to have both an environmental and genetic etiology. Established environmental factors include prolonged near work, education, and little outdoor exposure[12]. Genome-wide association studies (GWAS) and genetic linkage studies have identified multiple associated variants for refractive error[13–18]. The Consortium for Refractive Error and Myopia (CREAM) has reported numerous risk variants using large-scale, multiethnic datasets[19–22], explaining ~18% of phenotypic variance[22].

Despite estimates that 50% to 80% of refractive error variance is determined by genetic factors[23–26], much of the refractive error heritability remains unaccounted for[19,21]. Since GWAS are particularly designed to identify common variants, some of the missing heritability may lie with rare variants (minor allele frequency [MAF] ≤ 0.01), which may be highly penetrant and exert a large effect on the phenotype[27]. Gene-based association tests, such as burden-style tests[28,29], offer increased power to find rare variants not identified by GWAS.

This study performs a large-scale rare variant analysis on refractive error using multiethnic cohorts from CREAM. We used an initial discovery dataset consisting of over 13,000 Indo-Europeans and four replication datasets consisting of European ancestry Americans, European ancestry Australians, European ancestry Britons, and Eastern Asian ancestry Singaporeans. Gene-based tests were performed on each of the five cohorts and meta-analysis was performed subsequently. Pathway analysis was conducted on genome-wide significant genes and genes were prioritized based on annotation and biologic relevance to the trait.

## Results

**Overview of all analyses**. Across the three (i.e., VT, CMC and ACAT) multiethnic meta-analyses, the three Indo-European meta-analyses and the three EACC analyses, we identified a total of 129 unique genes that were significantly associated with the refractive error phenotype (Supplementary Data 3–5). We found no statistically significant difference in p-value or the number of unique genome-wide significant genes when adding the PRS as covariates.

**Multiethnic meta-analyses**. Forty-three genome-wide significant genes were found using EMMAX-VT (Fig. 1a), 11 genome-wide significant genes using the EMMAX-CMC (Fig. 1b), and 28 genome-wide significant genes using ACAT (Fig. 1c).

Sixty-eight unique genes were identified across the three tests (Fig. 2). Four genes were significant across all three tests - GDF15 (19p13.11), PDCD6IP (3p22.3), RRM2 (2p25.1), and ST6GAL-NAC5 (1p31.1). GDF15 (19p13.11) was one of the top two significant genes in all three approaches (EMMAX-VT $P = 5.12 \times 10^{-9}$, EMMAX-CMC $P = 1.12 \times 10^{-9}$, ACAT $P = 1.95 \times 10^{-9}$). GDF15, PDCD6IP, and RRM2 all replicated in at least one cohort; ST6GALNAC5 only appeared in IECC and thus could not be replicated.

Overall, using a replication p-value of 0.05, 25 genes were replicated using the EMMAX-VT approach: 11 in the ACAT approach and 4 in the EMMAX-CMC approach. Three genes — HCAR1, CCDC9, and NINJ2 — were replicated in more than one replication cohort, all in the EMMAX-VT approach. MRPS27 in EMMAX-VT (REHS and EPIC-Norfolk) and GDF15 in ACAT (IECC and REHS) had genome-wide significant p-values in two cohorts. If we use the more stringent replication threshold of $3.87 \times 10^{-4}$, then replications are observed for GDF15 (VT, CMC, ACAT) and MRPS27 (VT) with PDCD6IP (VT), NDC80 (VT) and LOXHD1 (ACAT) all having replication p-values very close to these thresholds. The list of all genome-wide significant genes for each test can be found in Supplementary Data 6–8, while the full results of all p-values can be found in Supplementary Data 9–11. Note that beta is provided for the individual CMC analyses and a direction for the individual VT analyses, as VT does not output a beta.

**Indo-European meta-analyses**. As it is possible that Eastern Asians differ in genetic risk factor profile from Indo-Europeans, we performed meta-analyses on the four Indo-European ancestry cohorts. Forty-nine genes were genome-wide significant in the EMMAX-VT approach (Fig. 3a), 13 genes in the EMMAX-CMC approach (Fig. 3b), and 29 genes in the ACAT approach (Fig. 3c). Four genes overlapped between all three tests — GDF15, PIK3CA, RRM2, and ST6GALNAC5 (Fig. 4). The signal at PIK3CA was unique to the Indo-European meta-analysis. GDF15 and RRM2 were both replicated in one cohort, while PIK3CA and ST6GALNAC5 only appeared in IECC.

Overall, 24 genes were replicated at $p = 0.05$ in EMMAX-VT, 8 genes in ACAT, and 4 genes in EMMAX-CMC. NINJ2 in the EMMAX-VT and STON1 and SND1 in EMMAX-CMC were replicated in multiple cohorts. The list of all genome-wide significant genes for each test can be found in Supplementary Data 12–14, while the full results of all p-values can be found in Supplementary Data 15–17.

**EACC analysis**. We also report the standalone results of EACC analysis. Thirty-one genome-wide significant genes were found in EACC using EMMAX-VT (Fig. 5a), 5 genome-wide significant genes using EMMAX-CMC (Fig. 5b), and 22 genome-wide significant genes using ACAT (Fig. 5c). GSTM5 (1p13.3) and WEE1 (11p15.4) overlapped in all three tests (Fig. 6). SERTAD3 (chromosome 19) and ZNF25 (chromosome 10) were genome-wide significant and only appeared in EACC, i.e., rare variants in these two genes did not exist in the other cohorts. 51 unique genome-wide significant genes were identified, 39 novel to the EACC analyses. The list of all genome-wide significant genes for each test can be found in Supplementary Data 18–20.

**Cohort unique genes**. In addition to the two genes in the EACC EMMAX-VT analysis, there were 6 significantly associated genes that only had rare variants within a single cohort; no other rare variants existed in the other cohorts for these genes. EDN3 and CHMP1B in the IECC EMMAX-VT analysis and PRLH in the IECC ACAT analysis. KLF1 appeared only in the EPIC-Norfolk cohort, in both the EMMAX-VT and EMMAX-CMC analyses. The list of cohort unique genes appears in Supplementary Data 21.

**Independent replication in UK Biobank**. We extracted the variants from the 129 significant unique genes and performed replication analyses in the UK Biobank. There were 7 genes with a $P < 0.05$ in EMMAX-CMC and 9 genes with a $P < 0.05$ in EMMAX-VT (Supplementary Data 22). P4HTM, CCDC170, and

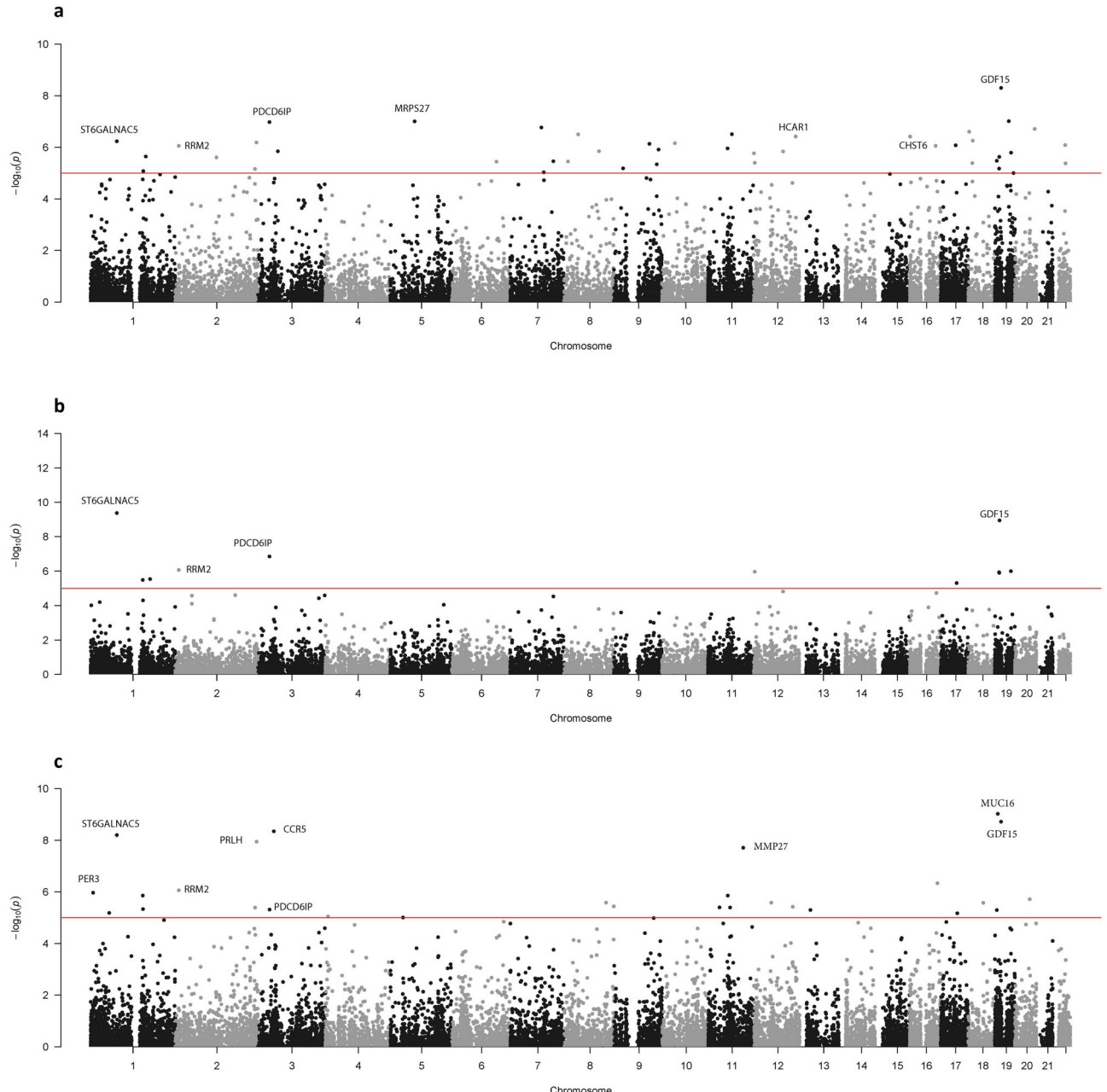

**Fig. 1 P-values of the multiethnic meta-analysis.** The gene-based p-values of the meta-analysis association study combining all five cohorts ($N = 27{,}006$) using the (**a**) EMMAX-VT test, (**b**) EMMAX-CMC test, and (**c**) ACAT. The line represents the genome-wide significant threshold of $1 \times 10^{-5}$. These plots are based on results in Supplementary Data 9–11, respectively.

*CPB1* were found in both analyses. *STON1* was also replicated in the UK Biobank analyses; this gene had a significant meta-analysis p-value in the EMMAX-CMC analysis. Interestingly, the p-value in all cohorts was <0.053.

**Pathway and expression analysis on all significant genes**. We performed IPA pathway analysis on the 129 unique genes. While this did not result in any genome-wide significant canonical pathways, the upstream regulators analysis identified over 172 associated transcription factors. The two highest were the cytokine *CSF2*, which is known to regulate neuroglia after retinal injuries[30], and the Transcription factor (TF) *MEF2C*, which is known to be expressed in the retina and controls photoreceptor gene expression[31] (Supplementary Data 23). The fourth ranked p-value was the Raf kinases, which are known to be involved in retinal development[32] and cell

survival;[33] the fifth ranked p-value was *TBX5*, which is expressed in the retina and involved in eye morphogenesis[34,35]. Causal network analysis identified 288 associated pathways (Supplementary Data 24), including the *TRPC5* pathway, which regulates axonal outgrowth in developing ganglion cells[36].

The top overall associated physiological system functions were organ morphology, organismal development and embryonic development, while the top molecular/cellular functions were cell cycle and cellular assembly/organization. Cancer and organismal injuries/abnormalities were the top overall associated phenotypes (Supplementary Data 25). Six genes were associated with ophthalmic phenotypes: *CHST6*, *GCNT2*, *P4HTHM*, *USH2A*, *GRHL2*, and *MAPT*.

FUMA analysis found that the top enriched tissues were heart, brain, muscle, and adipose tissue (Supplementary Fig. 1a). The top

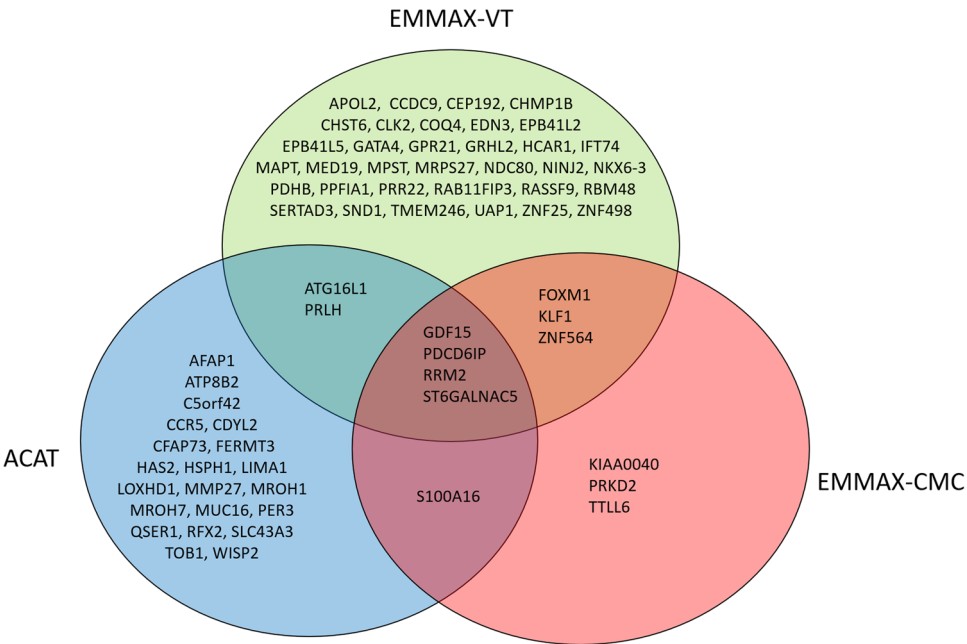

**Fig. 2 Overlap between three tests in the multiethnic meta-analysis.** A Venn diagram showing the overlap and unique significant genes in the multiethnic meta-analysis using the three different tests: EMMAX-VT (green), EMMAX-CMC (red), and ACAT (blue). These plots are based on results in Supplementary Data 6–8.

functional categories were cytoskeleton organization, cell cycle processes, mitotic nuclear division, and organelle organization (Supplementary Fig. 1b).

**Biological plausibility and prioritization of genes.** Of the 129 genome-wide significant genes from the six meta-analyses, 27.9% (36/129) have a known expression in human ocular tissue. 51.2% (66/129) of these genes showed evidence for a human ocular phenotype.

Seven genes had a biological plausibility score higher than 3 — *PER3* (internally replicated, expressed in ocular tissue and associated ocular phenotype, i.e., score of 5) and *PDCD6IP, MAPT, CHST6, GRHL2, USH2A*, and *P4HTM* (all with a score of 4). An additional 11 genes had a score of 3 — *GDF15, RRM2, HSPH1, TPR, KRT81, SPHK1, GSTM5, THSD7A, WEE1*, and *BUB1B* (Fig. 7). Table 1 provides the p-values and effect sizes of the prioritized genes. Detailed background for the prioritization of the genes can be found in Supplementary Data 26A–F. Supplementary Data 1 provides the p-values and effect sizes (when available) for each gene. Supplementary Data 27 provides the average SER for minor allele carriers versus non-carriers for each variant in these prioritized genes; please note that this table uses the single variant results which is restricted to MAC > = 3; some variants with MAC < 3 were used in the gene-based tests but will not be present in Supplementary Data 27. P-values and betas for each of the individual rare variants are also provided. In general, *PDCD6IP, MAPT*, and *USH2A* variants had the most negative average SER for carriers of the given rare variant (cases in the table), although genes *GRL2, CHST6, PDCD6IP*, and *USH2A* all had variants with high positive SER for rare variant carriers as well. Betas tended to conform with difference between rare variant carrier SER and SER in noncarriers (controls in the table), with many of the top variants having large betas. Perhaps the most interesting fact with respect to the betas is that most of the single variant betas tended to be positive and led to increased myopization (negative SER). However, there were still negative betas for some variants with more hyperopic mean SERs in

carriers versus non-carriers, particularly in the IECC and the genes *PDCD6IP* and *USH2A* across cohorts.

The highest overall biological plausibility score belonged to the circadian rhythm gene *PER3* (1p36). It was genome-wide significant in both the all cohorts ACAT and Indo-European only meta-analyses ($P = 1.08 \times 10^{-6}$ and $1.15 \times 10^{-6}$, respectively); it was genome-wide significant in REHS and replicated in IECC. CMC betas were 0.1666, 0.1574, −0.1976, −0.1102, −0.518 for IECC, EACC, BDES, EPIC-Norfolk, and REHS respectively; none of the CMC p-values were significant, however (Supplementary Data 28). Circadian rhythm genes have been shown to be associated with refractive error[22] and *PER3* is located near the site of a known myopia locus (MYP14) at which the causal gene has not been identified[37–39]. PER3 was expressed in ON and OFF bipolar cells. Defects in this gene are associated with familial advanced sleep phase syndrome (OMIM 616882) and may contribute to other circadian phenotypes by altering the sensitivity to light[40]. In defocus experiments in chicks using −15D lenses, PER3 expression decreased by −1.26-fold in the retina[41]. Further chick defocusing experiments, showed that PER3 expression in the retina varies under altered visual conditions[42]. Recently published data from the Raine Study suggest that falling asleep later was associated with a higher risk of myopia progression[43].

Five genes had a score just below *PER3*, including the apoptosis gene *PDCD6IP* (3p22.3). This gene was found to be genome-wide significant in all-cohorts meta-analyses using all three tests ($P = 1.07 \times 10^{-7}$, $1.45 \times 10^{-7}$, and $4.88 \times 10^{-6}$, respectively). Further *PDCD6IP* had a P of <0.006 in both the EACC and IECC cohorts and did not appear in the other cohorts. Both betas in the CMC test were negative and with a large effect size for IECC (beta = −2.5) (Supplementary Data 28). Most rare variants in this gene in the EACC and IECC samples result in mean SER's in carriers that were smaller (more negative) than in non-carriers, which meant that the CMC test would be powerful to detect this association (Supplementary Data 27). It is particularly interesting because *PDCD6IP* has two low single variant p-values in both IECC and EACC (0.00556 and 0.00548, respectively) and there

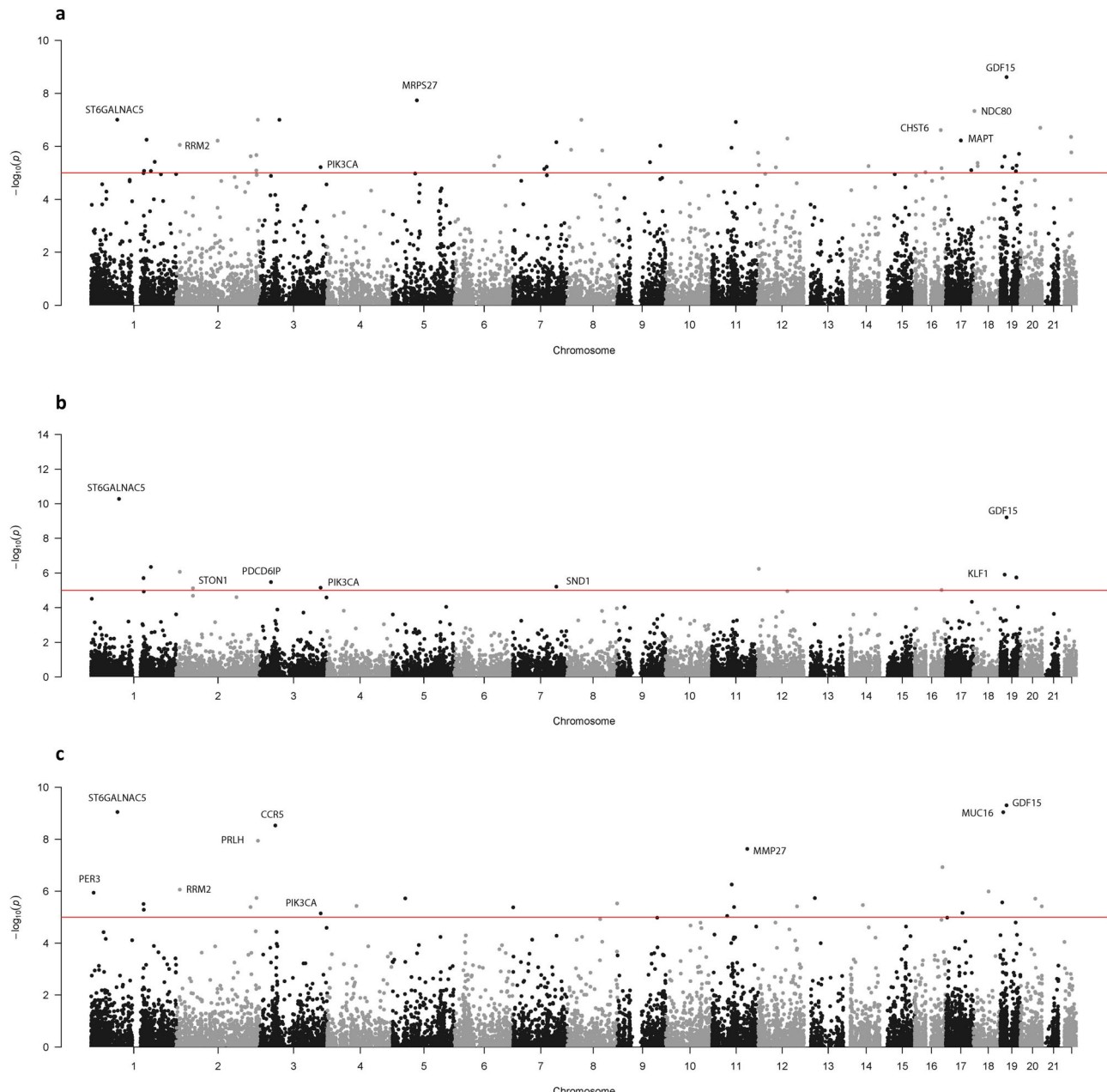

**Fig. 3 P-values of the Indo-European meta-analysis.** The gene-based p-values of the meta-analysis association study ($N = 22,139$) combining the four Indo-European derived cohorts using the (**a**) EMMAX-VT test, (**b**) EMMAX-CMC test, and (**c**) ACAT. The line represents the genome-wide significant threshold of $1 \times 10^{-5}$. These plots are based on results in Supplementary Data 15–17, respectively.

are no rare variants in this gene in any of the other cohorts. *PDCD6IP* is expressed in ganglion cells of peripheral retina and plays a role in programmed cell death in uveal melanoma[44] and may play a role in cornea lymphangiogenesis and vascular responses[45].

*MAPT* (17q21.32) encodes tau proteins responsible for stabilizing microtubules; it was found to be genome-wide significant in the all cohorts EMMAX-VT analysis ($P = 8.57 \times 10^{-7}$). It was genome-wide significant in REHS and replicated in EPIC-Norfolk. Betas from the CMC test were $-0.4342$, $0.3137$, $-0.4965$, $-0.171$, and $-0.8015$ for IECC, EACC, BDES, EPIC-Norfolk, and REHS respectively (Supplementary Data 28). Again, none of the CMC test p-values were significant. Abnormal MAPT was present in human glaucoma patients with uncontrolled intraocular pressure[46] Cowan et al. showed that *MAPT* was

expressed in several cell types in both the peripheral and foveal human retina: horizontal cells, rod bipolar cells, ON and OFF bipolar cells GLY and GABA amacrine cells and ganglion cells[47]. A knock-out mouse model showed decreased total retina thickness.

*CHST6* (16q23.1) was genome-wide significant in both the all cohorts and Indo-European only EMMAX-VT meta-analyses ($P = 8.99 \times 10^{-7}$ and $2.42 \times 10^{-7}$, respectively). The gene was genome-wide significant in IECC and replicated in BDES; it was also nearly replicated in EPIC-Norfolk. Though the CMC p-values were not significant, the beta for BDES was particularly large (0.95) (Supplementary Data 28). *CHST6* plays a role in maintaining corneal transparency. Mutations in this gene may result in macular corneal dystrophy (OMIM 217800), which is characterized by bilateral, progressive corneal opacification and a reduction of corneal sensitivity[48]. The mouse phenotype of a

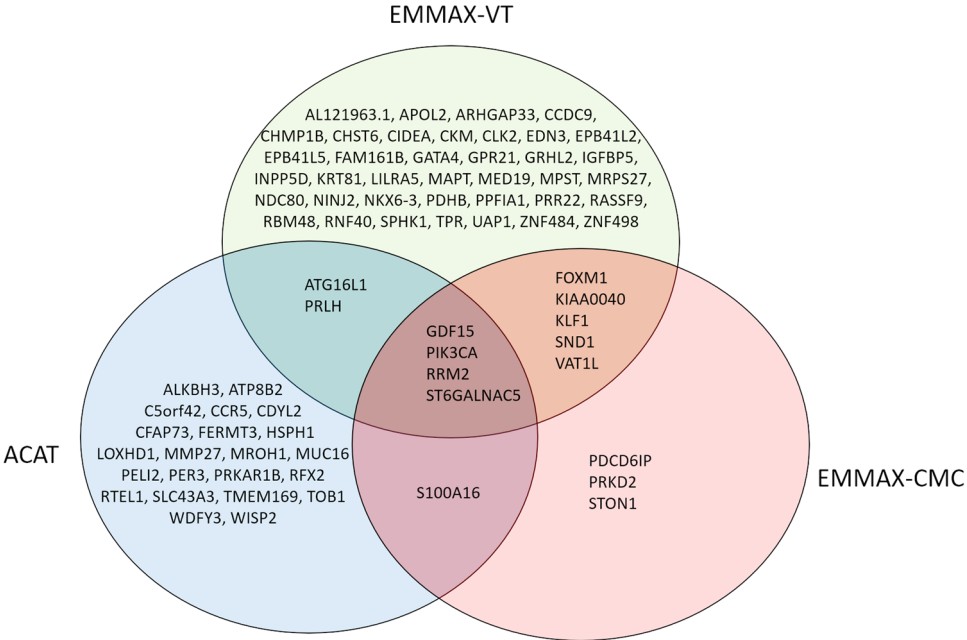

**Fig. 4 Overlap between three tests in the Indo-European meta-analysis.** A Venn diagram showing the overlap and unique significant genes in the Indo-European cohorts meta-analysis using the three different tests: EMMAX-VT (green), EMMAX-CMC (red), and ACAT (blue). These plots are based on results in Supplementary Data 12–14.

knock-out model corresponded to that of human, i.e., abnormal cornea morphology and decreased corneal (stroma) thickness. Since our reference expression database did not contain any corneal tissue, we couldn't score this category.

The transcription factor *GRHL2* (8q22.3) was genome-wide significant in the all cohorts EMMAX-VT meta-analysis ($P = 1.42 \times 10^{-6}$). It was genome-wide significant in REHS and replicated in IECC. Though the p-values for EMMAX-CMC were not significant, REHS had a large beta value of 0.87 (Supplementary Data 28). Mutations in GRHL2 may lead to posterior polymorphous corneal dystrophy[49] (OMIM 618031), characterized by a variable phenotype ranging from an irregular posterior corneal surface with occasional opacities, corneal edema, reduced visual acuity, secondary glaucoma, and corectopia.

The transmembrane prolyl hydroxylase *P4HTM* (3p21.31) was only genome-wide significant in EACC using EMMAX-VT ($P = 1.00 \times 10^{-7}$). However, this gene was replicated independently in the UKBB analysis. Betas for the non-significant EMMAX-CMC test were −0.1769, −02.025, 0.6106, −0.1632, −0.1177 for IECC, EACC, BDES, EPIC-Norfolk, and REHS respectively (Supplementary Data 28). *P4HTM* has been shown to be expressed in different ocular cells (including horizontal cells and bipolar cells). It is associated with HIDEA, a severe autosomal recessive disorder that is characterized by multiple symptoms, including eye abnormalities[50] (OMIM 618493) and knock-out mice models have shown abnormal eye morphology[51].

The membrane gene *USH2A* (1q41) was genome-wide significant in the EACC ACAT analysis ($P = 7.55 \times 10^{-9}$). The EMMAX-CMC tests were not significant which is reflected in the betas which were all quite small except for 0.82 in the BDES sample (Supplementary Data 28). This reflects the wide variation in effect on SER exhibited by different rare variants in this gene, with some individual variants leading to much more myopic mean SER's in carriers compared to non-carriers while other rare variants led to more hyperopic mean SERs in carriers compared to non-carriers. (Supplementary Data 27). *USH2A* is well known to cause both Usher syndrome, which includes retinitis pigmentosa (RP) and mild to moderate hearing loss, as well as

RP without hearing loss[52]. It is known to be expressed in the retina[53] and has been recently shown to be associated with high myopia[54]

**Pathway and expression analysis on top prioritized genes**. We ran the IPA and FUMA analyses on the seven top prioritized genes. IPA did not identify any canonical pathways as significant; the only pathway shared across the genes was the 14-3-3-mediated signaling pathway (*MAPT* and *PDCD6IP*). The 14-3-3 proteins are a diverse group of signaling proteins.

Upstream regulator analysis found several transcription regulators of at least two genes include *NKX2-1* (*GRHL2* and *MAPT*), *PSEN1* (*MAPT* and *PER3*), and *SIRT1* (*MAPT* and *PDCD6IP*) (Supplementary Data 29). In the causal network analysis, the master regulator with the highest p-value covering multiple genes was the cytokine macrophage migration inhibitory factor (*MIF*) (Supplementary Data 29), which covered five genes. Interestingly, *MIF* is an essential factor in the development of zebrafish eyes[55] and has been found to be a potential regulator of diabetic retinopathy[56]. *MIF* inhibitors may also be protective to photoreceptors[57]. Causal network analysis can be found in Supplementary Data 30 and the top functional analysis for disease result was hereditary eye disease (Supplementary Data 31). FUMA showed the top tissue expression occurred in the small intestinal terminal ileum, skeletal muscle, and the brain cortex; the latter being probably the best proxy for eye tissue (Supplementary Fig. 2a). A heat map of the expression of the seven genes across all GTEx tissues is given in Supplementary Fig. 2b).

**Potential causal variants in the prioritized genes**. We used annotation from wANNOVAR to identify potential causal variants within the top genes identified by the prioritization method (Table 2). For the two prioritized genes that were significant in the ACAT analyses, we were able to look at single variant p-values in addition to annotation to determine potential causal variants. There were three good candidate variants in *PDCD6IP*, which was

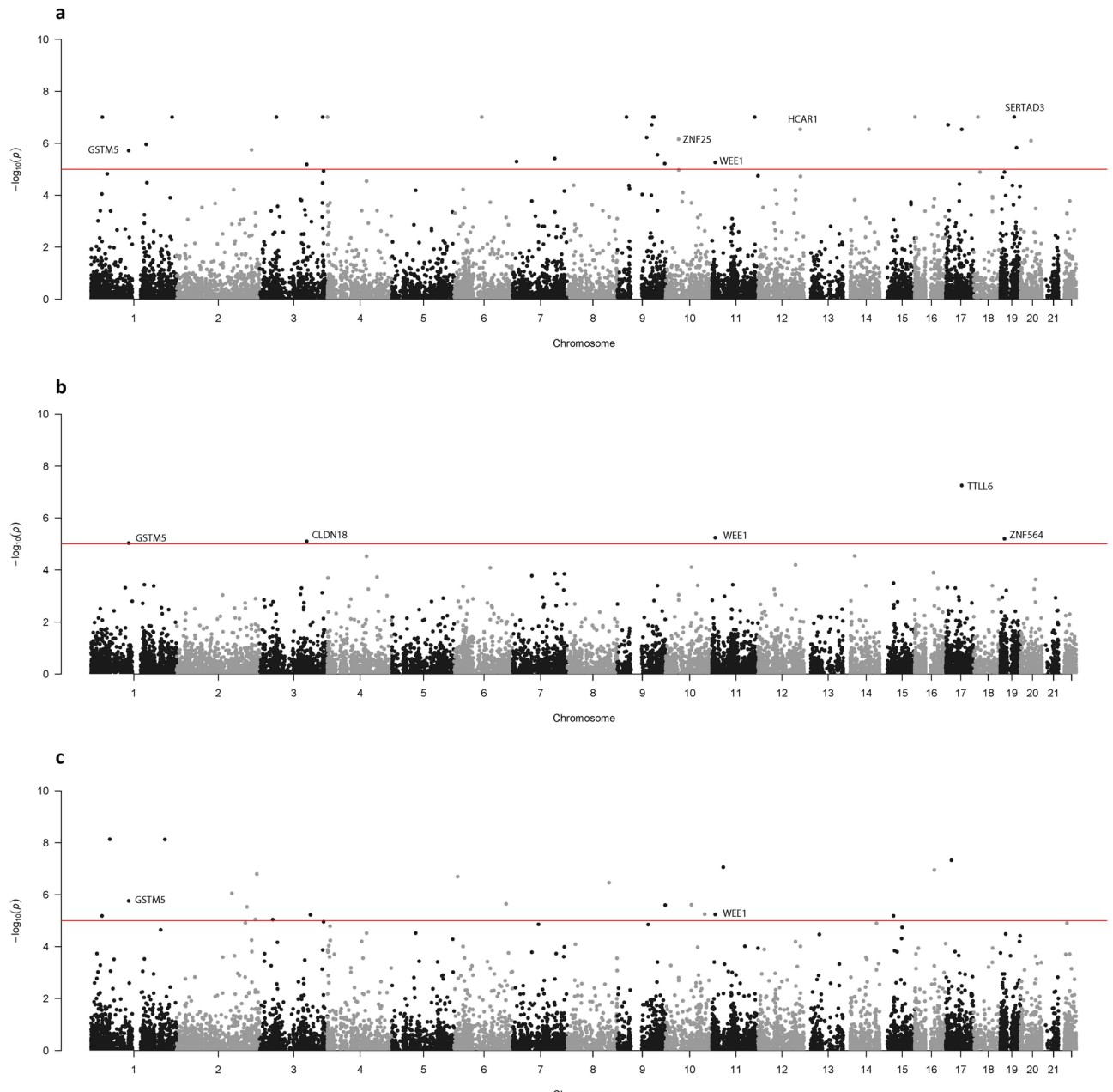

**Fig. 5 P-values of the analysis using the Eastern Asian EACC only.** The gene-based *p*-values of the EACC association analysis (*N* = 4867) using the (**a**) EMMAX-VT test, (**b**) EMMAX-CMC test, and (**c**) ACAT. The line represents the genome-wide significant threshold of 1 ×10$^{-5}$. These plots are based on results in Supplementary Data 9–11 respectively.

genome-wide significant in IECC and replicated in EACC. rs199990824 (3:3879764) appeared in the EACC only, was predicted to be damaging by SIFT and MutationTaster, and had a CADD score of 26. The minor allele of rs199990824 appeared in 37 carriers (all heterozygotes) with an average SER of −2.04 D (SD = 3.29) compared to the non-carrier average of −0.44 D (SD = 2.27) and the overall cohort average of −0.45 D (SD = 2.28); the single variant *P* was 0.000183. In the IECC, the best potential causal variant was rs62620697 (3: 33905532), which was predicted damaging by MutationTaster, had a CADD score of 23.8, and had a low single variant *p*-value of 0.002632. Carriers (*N* = 9) of rs62620697 had an average SER of −2.17D (SD = 6.87) compared to that of non-carriers with an average SER of 0.20 (SD = 2.27). rs145293758 also had a low *p*-value (0.000311) but was not predicted damaging.

Potential candidate variants were also identified in *PER3*, which was genome-wide significant in REHS and replicated in IECC. The REHS signal was primarily driven by two variants - rs147327372 and rs144178755, which had single variant *p*-values of $1.72 \times 10^{-8}$ and 0.004953, respectively. However, neither variant was predicted to be damaging by the prediction algorithms nor appeared in the other European cohorts and were not significant individually, although rs147327372 did have a *p*-value of 0.046 in EPIC-Norfolk in the single variant tests.

The signals in the other four genes, identified primarily by the two burden-style tests, were driven by a cumulative effect of several variants. In this case, we relied primarily on annotation and reported variants that were generally agreed upon by multiple prediction programs. Five good candidate variants were located in *MAPT*: rs139796158 (17:44055786), rs76375268 (17:44060807),

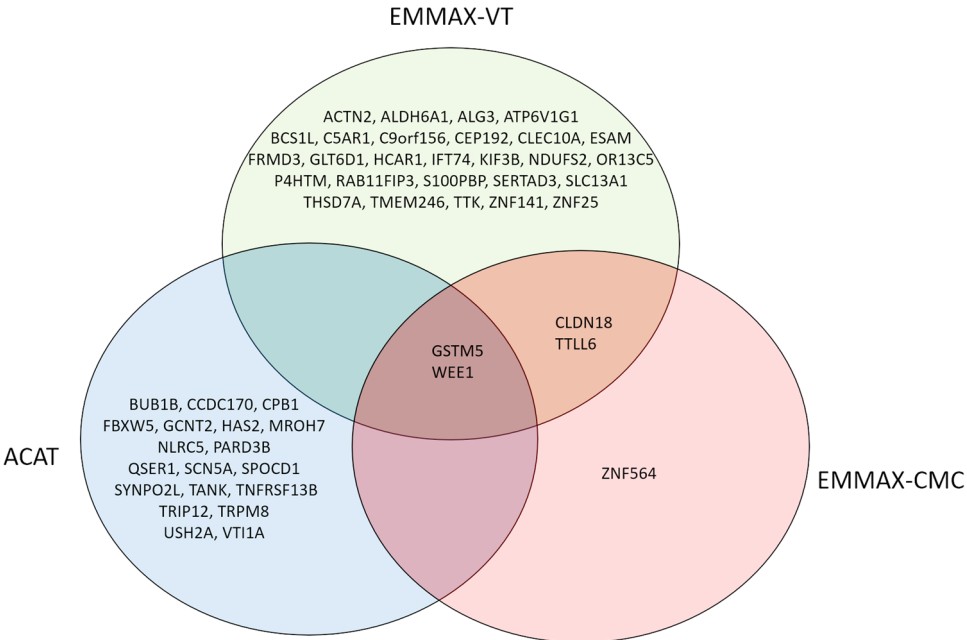

**Fig. 6 Overlap between three tests in the Eastern Asian EACC analysis.** A Venn diagram showing the overlap and unique significant genes in the EACC analysis using the three different tests: EMMAX-VT (green), EMMAX-CMC (red), and ACAT (blue). These plots are based on results in Supplementary Data 18–20.

| Gene | | | Meta-analysis | Individual populations | | | | | Internal replication | | Expression | Biology | | | GWAS | Total | Drug |
|---|---|---|---|---|---|---|---|---|---|---|---|---|---|---|---|---|---|
| #Chr | Pos | Gene | Meta-pvalue | Asian-NT | EPIC | BDES | Indo-European | Raine | one cohort <=10-5 and the other p<0.05 | ACAT-all | Total (any of 4 models) | Ocular phenotype in mice | Ocular phenotype in humans | GWAS overlap (Hysi et al.) | Range 0-6 | |
| 3 | 33877626 | PDCD6IP | 1.07E-07 | 0.00041 | NA | NA | 0.000013 | NA | 1 | 1 | 1 | 0 | 1 | 0 | 4 | X |
| 17 | 44039717 | MAPT | 8.57E-07 | 0.19 | 0.039 | 0.44 | 0.17 | 1E-07 | 1 | | 1 | 1 | 1 | 0 | 4 | X |
| 16 | 75512672 | CHST6 | 8.99E-07 | 0.56 | 0.092 | 0.0095 | 2E-07 | 0.6 | 1 | | 0 | 1 | 2 | 0 | 4 | X |
| 8 | 1.03E+08 | GRHL2 | 1.42E-06 | NA | 0.38 | 0.86 | 0.0082 | 3E-07 | 1 | | 0 | 1 | 2 | 0 | 4 | X |
| 19 | 18497141 | GDF15 | 5.12E-09 | 0.2 | 0.99 | NA | 2E-07 | 0.000034 | 1 | 1 | 0 | 0 | 1 | 0 | 3 | X |
| 2 | 10262920 | RRM2 | 8.81E-07 | NA | NA | 0.27 | 0.0088 | 1.8E-06 | 1 | 1 | 0 | 0 | 1 | 0 | 3 | X |

**Fig. 7 Prioritization of top genes from all 129 genome-wide significant genes.** The top genes ranked by our prioritization schema. The figure contains the chromosome, basepair position, gene name, as well as the meta-analysis *p*-value and the individual cohort *p*-values for each gene. It also contains which test the given significant meta-analysis *p*-value refers to, and how many times the gene replicated in our internal analyses. Finally, it contains information regarding gene expression, whether the gene has a known ocular phenotype in mice or humans, overlap with the GWAS performed by Hysi et al., and the final overall prioritization score. This figure is based on results shown in Supplementary Data 26.

rs63750072 (17:44060859), rs143956882 (17:44067341) and rs63750191 (17:44101481). All these variants were nonsynonymous variants and predicted damaging by three of the four databases (except for rs76375268, which was predicted damaging by two). rs139796158, rs143956882, and rs63750191 all had CADD scores >26. In *CHST6*, the best candidate variant was the missense variant rs140699573 (16:75512734). It was predicted damaging by SIFT, PolyPhen2, MutationTaster, and FATHMM and has a CADD score of 27.4. In *GRHL2*, the best candidate variant was rs142411476 (8:102570910). It was predicted damaging by two databases and had a CADD score of 22. In *P4HTM*, two variants of interest were identified: rs140290144 (3:49002551) and rs144279528 (3:49043292). These variants were predicted damaging by MutationTaster and had CADD scores of 22.1 and 27.3, respectively. Finally, in *USH2A*, three variants (rs554957414 (1:216138793), rs148135241 (1:216373416), and rs201527662 (1:216419934) were all predicted

damaging by the five prediction algorithms and had CADD scores above 22.

**Structural analysis of prioritized candidate proteins**. In addition to the annotation, we also performed protein structural modeling of all coding variants within the prioritized genes (98 variants across 6 genes/proteins) and calculated free energy difference (ΔΔG) between wildtype and mutant proteins (Supplementary Data 32); positive ΔΔG indicates a shift from a more stable to a less stable isoform. More detailed information on the structural analysis can be found in the Supplemental Methods.

In *PDCD6IP*, both rs145293758 (3:33905587) and rs200697599 (3: 33840234) were predicted to be highly destabilizing to protein structure (Supplementary Fig. 3a). The variant rs145293758 leads to replacement of a proline (Pro737) for an asparagine near phosphorylation sites in the protein's self-associating domain,

**Table 1 P-values and effect sizes of prioritized genes.**

| | Meta-analysis | | | Discovery set | | | Replication sets | | | | | | | | | | | | |
| | Multiethnic P-values | | | IECC P-values | | | EACC P-values | | | BDES P-values | | | EPIC-Norfolk P-values | | | REHS P-values | | |
| Gene | CMC | VT | ACAT | CMC (beta) | VT | ACAT | CMC (beta) | VT | ACAT | CMC (beta) | VT | ACAT | CMC (beta) | VT | ACAT | CMC (beta) | VT | ACAT |
|---|---|---|---|---|---|---|---|---|---|---|---|---|---|---|---|---|---|---|
| PDCD6IP | 2.45e−7 | 1e−7 | 4.9e−6 | 3.4e−6 (0.0095) | 1.3e−5 | 5.6e−4 | 0.002 (−0.76) | 4.1e−4 | 5.5e−4 | NA | NA | NA | NA | NA | NA | NA | NA | NA |
| PER3 | 0.08 | 0.03 | 1e−6 | 0.14 (0.17) | 0.04 | 0.05 | 0.27 (0.16) | 0.55 | 0.12 | 0.51 (−0.2) | 0.42 | 0.48 | 0.43 (−0.11) | 0.11 | 0.22 | 0.02 (−0.52) | 0.04 | 1.2e−7 |
| USH2A | 0.44 | 0.90 | 1.3e−5 | 0.82 (0.01) | 0.96 | 0.67 | 0.38 (−0.08) | 0.75 | 7.6e−9 | 0.78 (−0.05) | 0.72 | 0.35 | 0.33 (0.08) | 0.90 | 0.98 | 0.08 (0.22) | 0.19 | 0.81 |
| MAPT | 0.02 | 8.6e−7 | 0.57 | 0.02 (−0.43) | 0.17 | 0.18 | 0.15 (0.31) | 0.19 | 0.37 | 0.43 (−0.5) | 0.44 | 0.24 | 0.48 (−0.17) | 0.04 | 0.87 | 0.01 (−0.8) | 1e−7 | 0.94 |
| GRHL2 | 0.47 | 1.4e−6 | 0.06 | 0.31 (0.33) | 0.008 | 0.08 | NA | NA | NA | 0.87 (−0.19) | 0.86 | 0.08 | 0.39 (−0.55) | 0.38 | 0.39 | 0.21 (0.87) | 3e−7 | 0.21 |
| CHST6 | 0.19 | 9e−7 | 0.06 | 0.58 (−0.09) | 2e−7 | 0.40 | 0.38 (0.21) | 0.56 | 0.58 | 0.05 (0.96) | 0.009 | 0.01 | 0.27 (0.27) | 0.09 | 0.16 | 0.35 (0.42) | 0.6 | 0.42 |
| P4HTM | 0.09 | 1.7e−5 | 0.16 | 0.39 (0.01) | 0.24 | 0.10 | 0.005 (−2.02) | 1e−7 | 0.14 | 0.34 (0.61) | 0.29 | 0.19 | 0.49 (−0.16) | 0.51 | 0.54 | 0.83 (−0.12) | 0.56 | 0.57 |

Legend: The summary statistics from our prioritized genes. The p-values for the CMC, VT, and ACAT analyses and betas for the CMC analyses for the meta-analysis and the five individual cohorts. Note that for the CMC meta-analysis, no beta is provided because Fisher's method does not provide an effect size. Sample sizes for each of the cohorts are as follows: multiethnic meta-analysis (N = 27,006), IECC (N = 13,097), EACC (N = 4867), BDES (N = 1740), EPIC-Norfolk (N = 6282), and REHS (N = 1020).

which could disrupt phosphorylation. rs200697599 (Ile5) and rs199990824 (Asp376; 3:33879764) result in changes to the protein's BRO1 domain, which is involved in endosomal targeting. The isoleucine to serine mutation at rs200697599 could introduce a phosphorylation site at the N-terminus while the asparagine to aspartic acid mutation at rs199990824 could disrupt hydrogen bonds. Recall that both rs145293758 and rs199990824 were identified as potential causal variants for refractive error in IECC and EACC, respectively, based on their annotation, and single variant p-values (Supplementary Data 27).

For *PER3*, several variants may affect structure, including rs140974114, which results a serine (Ser751) to aspartic acid substitution at the protein's nuclear localization signal and could disrupt hydrogen bonds and rs200140283, which results in an alanine (Ala681) to glycine substitution in the CSNK1E binding domain. Further potential disruptions occur at rs139315125 (His416), which takes place in the nuclear export signal 3 and rs77418803 (Ser919), which occurs near the nuclear export signal 2. The model is provided in Supplementary Fig. 3b.

Of the variants in *MAPT*, two were predicted to be destabilizing (rs76375268 at Gly213 and rs63750191 at Gln741) (Supplementary Fig. 3c). Further, rs73314997 (Ser318) and rs143956882 (Ser427) are located near known pathogenic mutations for frontotemporal dementia and Pick disease of the brain, respectively.

Three variants on the luminal domain of CHST6 were found to have a mild effect on protein stability. Two of these variants (rs201349198 at Ala326 and rs140699573 at Gln331) are positioned near variants known to cause macular corneal dystrophy (MCD) near the C-terminus. This suggests the C-terminus is sensitive to mutations enabling interference with keratan sulfation, which could cause a loss of function that can lead to a milder disease phenotype such as refractive error. The model can be found in Supplementary Fig. 4a.

In *GRHL*2, variants were only predicted to have a mild effect on protein structure and were not located near known pathogenic variants (Supplementary Fig. 4b).

For *P4HTM*, rs140290144 is predicted to be moderately destabilizing (Supplementary Fig. 4c). It substitutes a valine for a buried isoleucine (Ile227) between two calcium binding sites; potential disruption of these calcium binding sites can result in loss of function. Similarly, rs144279528 occurs in the Fe-dependent 2-OG dioxygenase domain close to an iron binding residue. Substitution of asparagine from the wildtype aspartic acid (Asp386) could have an impact on iron binding by introducing a glycosylation (due to location on protein surface) or disruption of hydrogen bonding.

Of particular interest in the protein modeling was that of usherin (*USH2A*), the known retinitis pigmentosa gene. Five variants were predicted to be highly destabilizing, particularly rs554957414 with a ΔΔG value of 99.19 kcal/mol). Three of these variants, including rs554957414 (Pro2329), result in the loss of proline and the loss of that ring structure could cause an increase in conformational flexibility and account for such high destabilization predictions (Supplementary Fig. 5). Further, a mutation at rs201527662 (Cys934) results in the replacement of cysteine with tryptophan and will disrupt a disulfide bond between two cysteines.

We also compared the ΔΔG of these five candidate variants with the ΔΔG of all *USH2A* ClinVar ($n = 63$) and gnomAD ($n = 1870$) variants using the Wilcoxon rank sum test. A significant difference between the ClinVar variants and gnomAD variants was found ($P = 0.0008$) and the ΔΔG values of our candidate variants was much more similar to the known pathogenic variants than the putatively benign GnomAD variants (Supplementary Fig. 6).

**Table 2 Potential missense causal variants in prioritized genes.**

| CHR | BP | rs ID | Gene | AA change | MAF | SIFT | PolyPhen2 | MT | FATHMM | CADD |
|---|---|---|---|---|---|---|---|---|---|---|
| 1 | 7879401 | rs147327372 | PER3 | Thr519Ala | 0.002 | T | B | N | T | 0.01 |
| 1 | 7890153 | rs144178755 | PER3 | Thr1040Asn | 0.001 | D | B | N | T | 0.962 |
| 1 | 216138793 | rs554957414 | USH2A | Pro2329Leu | 2e−6 | D | D | D | D | 29.1 |
| 1 | 216373416 | rs148135241 | USH2A | Ser1122Pro | 0.004 | D | D | D | D | 22.8 |
| 1 | 216419934 | rs201527662 | USH2A | Cys934Trp | 0.0002 | D | D | D | D | 36 |
| 3 | 33840234 | rs200697599 | PDCD6IP | Ile5Ser | 0.0007 | D | D | D | T | 32 |
| 3 | 33879764 | rs199990824 | PDCD6IP | Asp376Asn | 4e−6 | D | B | D | T | 26 |
| 3 | 33905532 | rs62620697 | PDCD6IP | Ala719Thr | 4e−6 | T | B | D | T | 23.8 |
| 3 | 33905587 | rs145293758 | PDCD6IP | Pro737Arg | 0.001 | T | B | N | T | 20.2 |
| 3 | 49039984 | rs140290144 | P4HTM | Ile227Val | 0.006 | T | B | D | T | 22.1 |
| 3 | 49043292 | rs144279528 | P4HTM | Asp386Asn | 8e−5 | T | B | D | T | 27.3 |
| 8 | 102570910 | rs142411476 | GRHL2 | Arg183Gln | 0.0002 | T | D | D | T | 22 |
| 16 | 75512734 | rs140699573 | CHST6 | Gln331His | 4e−6 | D | D | D | D | 27.4 |
| 17 | 44055786 | rs139796158 | MAPT | Ala118Gly | 6e−5 | D | D | D | T | 26.4 |
| 17 | 44060807 | rs76375268 | MAPT | Gly213Arg | 0.004 | D | D | N | T | 11.71 |
| 17 | 44060859 | rs63750072 | MAPT | Gln230Arg | 0.04 | D | D | D | T | 4.652 |
| 17 | 44067341 | rs143956882 | MAPT | Ser427Phe | 0.001 | D | D | D | T | 28.5 |
| 17 | 44101481 | rs63750191 | MAPT | Gln741Lys | 3e−5 | D | D | D | T | 27.5 |

Legend: The best potential missense causal variants in our top prioritized genes. The headers represent: CHR = chromosome, BP = physical position in basepairs (hg19), Gene = gene location, AA change = amino acid change caused by mutation, MAF = minor allele frequency of the variant obtained from gnomAD, SIFT = pathogenicity prediction from SIFT (where T = tolerated and D = damaging), PolyPhen2 = pathogenicity prediction from PolyPhen2 (where B = benign and D = damaging), MT = pathogenicity prediction from MutationTaster (where N = neutral and D = damaging), FATHMM = pathogenicity prediction from FATHMM (where T = tolerated and D = damaging), CADD = CADD phred score.

**Potential causal variants in other genome-wide significant genes**. We also identified variants within the other 122 genome-wide significant genes that had a high potential to be damaging. This included 25 variants across the five cohorts; the results are found in Supplementary Data 33. Like our prioritized genes, we also performed protein modeling on these variants (Supplementary Data 34).

Notable findings from the structural analysis include a valine to phenylalanine substitution (Val105) that would disrupt a helix in ALG3, which has been implicated in congenital disorders of glycosylation that have ocular phenotypes[58] (Supplementary Fig. 7a). We also identified multiple glycine substitutions in TNFRSF13B in areas associated with heparan sulfate – glycosaminoglycan biosynthesis; heparan sulfate has been shown to play a role in eye pathologies[59] (Supplementary Fig. 7b).

## Discussion

In this large scale, gene-based analysis of rare variants in refractive error, 129 associated genes were identified. Though many of the genes were associated with eye conditions or ocular development, only ten genes had previously been identified with refractive error or myopia: six with myopia including two with high myopia — USH2A and GDF15[54,60] — and ten with refractive error. Pathway analysis revealed that 59 of these genes were involved in cell cycle, organ morphology, and embryonic development and 21 of these genes had upstream regulators that were directly involved in retinal development or eye morphogenesis. Given the substantial level of missing heritability still present within the refractive error, it is likely that at least some of this heritability is explained by rare variants within these genes. The fact that the significance of these genes and the explained variance of refractive error due to these genes did not significantly change after inclusion of GRS in the analysis, suggests that these association signals are independent from the effects of known common refractive error risk variants.

This large scale meta-analysis used gene-based tests for rare variants in refractive error, and was undertaken to identify rare variants that may be partially responsible for missing heritability, particularly within the CREAM data set[21]. The CREAM data set

is well-suited for this type of rare variant analysis. First, we were able to combine many smaller cohorts into two mega-analyses – IECC ($N = 11,505$) and EACC ($N = 4867$). These meta-analyses greatly boosted power to detect variants with a MAF ≤ 0.01 and allowed more rare variants to be combined into a single, gene-based marker. In addition, we had three cohorts >1000 subjects to observe replication and perform the combined meta-analyses. Genes identified in this study were done so across a very large pool of subjects, lowering the potential for type I error.

The multiethnic composition of this dataset also allowed for observation both across and within ethnicities. We have delineated how rare variants in some genes were found only in Indo-Europeans and others in Eastern Asians, as well as some that cut across the ethnic divide. Thus, we were able to identify risk genes that might contain rare variants that affect SER within a particular population (such as ST6GALNAC5 in IECC), or more universally, like PDCD6IP.

PER3, PDCD6IP, MAPT, CHST6, P4HTM, USH2A, and GRHL2 are good candidate genes, all known to be associated with ocular abnormalities. PER3 is a circadian rhythm gene; circadian rhythm is associated with refractive error[22]. PDCD6IP and MAPT are both expressed in the retina while CHST6, and GRHL2 are both involved in corneal dystrophy[49,61]. P4HTM affects eye morphology in mice knockouts;[55] it is also notable for being replicated in the UKBB analysis. USH2A is expressed in the retina and is a known RP gene[52,53].

Five of these prioritized genes were found to be regulated by the cytokine MIF, which has been shown to regulate zebrafish eye development[55] and have protective effects for photoreceptors[57]. More work on the MIF network with respect to refractive error is needed. We were further able to identify potential causal variants in these prioritized genes and, using structural analysis, were even able to determine the effect on protein stability.

STON1, C5AR1, and WDFY3 were all replicated in UKBB. C5AR1 is expressed in retinal Müller cells, which are known to play a role in retinal disease[62]. STON1 is associated with AMD[63] while WDFY3 is associated with inherited retinal dystrophies[64]. Other potential interesting candidates include GDF15, which was a top significant gene across all four meta-analyses, and has been found to be significantly overexpressed in highly myopic eyes[60]

and patients with vitreoretinal disorders[65] and may also be a potential molecular marker of neurodegeneration in glaucoma[66], and *MRPS27*. This gene was genome-wide significant in the meta-analysis and in two individual cohorts, REHS and EPIC-Norfolk. While *MRPS27* is not known to be associated with eye disease, a common variant in this gene was found to be genome-wide significant in the GWAS meta-analysis of refractive error conducted by Hysi et al.[22] Other candidate genes with known links to eye disease/functions include *HCAR1* with glaucoma[67,68] and *EPB41L2* with a potential role in phototransduction[69].

One final interesting set of genes was those that were genome-wide significant within a single cohort. This implies that there may be rare risk variants unique to a certain population that are fixed in other populations. This includes *ST6GALNAC5*, which was genome-wide significant in IECC in both EMMAX-VT and ACAT ($P = 5.84 \times 10^{-7}$, $9.03 \times 10^{-10}$). This gene catalyzes the transfer of sialic acid; polysialic acid has been shown to prevent vascular damage in retina[70] and to stimulate the generation of new rods in the retinas of developing zebrafish[71]. Other interesting significant genes unique to a single cohort included *SERTAD3* in EACC, which is overexpressed in retinoblastoma[72] and *KLF1* in EPIC-Norfolk, which may be expressed in the eye[73]. We also note that gene-based analyses for refractive error had been previously performed in BDES[74]. Of the five significant genes from that analysis, two were replicated at $P \leq 0.05$ — *PTCHD2* and *CRISP3*. *PTCHD2* is located near the known myopia locus *MYP14* on 1p36.22[39,75] and *CRISP3* is expressed in the retina[74,76].

This study used multiple tests (EMMAX-VT, EMMAX-CMC and ACAT) to identify significant genes and looked at overlap to find more robust signals. By using multiple tests that differ slightly in design, we were able to cast a wider net in our search. The ACAT test was particularly useful for identifying potential causal variants within a candidate gene, as it allowed us to observe which variants had significant single variant *p*-values. This enabled us to zero in on potential causal variants in genes like *PDCD6IP* and *PER3*, though we note that highlighting any potential causal variants are speculative at this point. We also felt it prudent to not give more weight to the result of one test over another and instead take the largest number of unique, significant genes since this was a discovery study, though we did try to give more weight to the genes that were identified by all three tests, such as *PDCD6IP*.

We note that the three tests did not always agree, though the two burden-style tests agreed more often than ACAT. This is not surprising given the different nature of the tests. Both EMMAX-VT and EMMAX-CMC were burden-style tests that create a new, gene-based marker on which the *p*-value is calculated. The ACAT test was an aggregation-style test created from single variant *p*-values that does not create a new gene-based marker[77]. This is a critical distinction; it means that the markers analyzed in the burden-style tests and the ACAT tests are different. The ACAT analyses may have been slightly underpowered with respect to the burden-style tests, as we used a minimum allele count of three in our analyses. For EMMAX-VT and EMMAX-CMC this was calculated across all variants within a gene and for ACAT at each individual variant, which resulted in certain variants being removed from the ACAT analysis that were present in the burden style analyses. Therefore, genes present in all three analyses indicate a more robust association with refractive error.

Since this is an exome microarray study, there were still large portions of the genome that would not have been covered in this work. Thus, there are almost certainly additional rare risk variants for refractive error in these cohorts that were not genotyped in this study. The goal of this discovery study was to provide an initial starting point for further analysis; we plan whole genome sequencing on high-risk individuals identified by this study. These non-genotyped variants could explain why we did not see replication with previous refractive error GWAS findings[21,22]. Some of the genes identified in the common variant GWAS may have included rare risk variants that were specific to a particular population that was not used in this study.

Another challenge is that due to the gene-based nature of this work, it is critical to remember that the gene-based markers across the cohorts are often made up of different variants. This means that the gene-based marker for gene A in IECC might be made up of three variants, and in REHS might be made up of seven variants, two of which are shared across the two cohorts. This means that it was possible that some cohorts may have had association tests that were less significant because of inclusion of non-significant rare variants that did not appear in other cohorts.

We also note that this was an exploratory analysis to determine candidate genes, and one of our goals was to cast a wide net to capture potential candidates. Therefore, we chose a more liberal replication significance threshold, which may allow for potential type I errors but would also ensure that a good candidate gene would not be missed or because functional rare variants did not appear in that cohort.

We also note that while we did utilize eye expression data in this study, we were limited to expression from retinal tissue only. We are actively seeking expression data from additional eye tissue, particularly corneal and scleral tissue, to further prioritize these genes.

This work identified 129 genome-wide significant genes for refractive error using the gene-based rare variant approach. Most of these genes are novel for association with refractive error but many have associations with other ocular abnormalities. This is the largest gene-based study of rare variants performed on refractive error. The fact that we found over 100 significant genes shows that rare variants (MAF ≤ 0.01) do account for some of the missing refractive error heritability not identified in the common variant GWAS. We were able to prioritize seven of these genes as our best candidate genes for causality based on biological function – *PDCD6IP*, *MAPT*, *CHST6*, *GRHL2*, *USH2A*, *P4HTM*, and *PER3* –as well as *GDF15* and *MRPS27* based on the strength of association. Validation studies, including replication within additional cohorts, are planned to identify the best candidates for functional studies to unravel the pathophysiology of refractive error and myopia. We also plan further analysis with the conversion of our quantitative refractive error phenotype to binary phenotypes to test for association with myopia, hyperopia, and astigmatism.

## Methods

**Cohort details, genotyping and joint recalling of exome array data.** Fourteen population-based CREAM cohorts that had exome chip genotypes on individuals with refractive error measurements were used in this study. These 14 cohorts were: Singapore Chinese Eye Study (SCES), Singapore Malay Eye Study (SiMES), Singapore Indian Eye Study (SINDI), Age Related Eye Study (AREDS), Rotterdam Study I (RSI), Erasmus Rucphen Family (ERF), Raine Eye Health Study (REHS) of the Raine Study, Beaver Dam Eye Study (BDES), Estonian Genome Center for the University of Tartu (EGCUT), Finnish Twin Study on Aging (FITSA), Ogliastra, Croatia-Korcula, TwinsUK, and EPIC-Norfolk. Each individual cohort is described in further detail in Supplementary Note 1. All studies were performed in accordance with the Declaration of Helsinki and approved by the institutional review boards of the participating institutions. All participants provided written informed consent. The Institutional Review Board of the National Institutes of Health (NIH) determined that the analyses of deidentified data performed in the current study and the meta-analysis qualified as "not human subjects research" and did not require specific protocol approval. The study was performed under guidelines agreed to in Data Use Agreements between the individual participating studies and the NIH and the Erasmus Medical Center where these analyses took place.

Thirteen cohorts had been genotyped on the Illumina HumanExome-12 v 1.0 or v 1.1, or the Illumina HumanCoreExome-12 v1.0; EPIC-Norfolk was genotyped on Affymetrix UK BioBank Axiom Array. The 13 cohorts on the Illumina arrays were jointly recalled to obtain a larger sample size of rare variants (here defined as variants with a MAF ≤ 0.01), as recalling genotypes simultaneously across all

samples increases the ability to call rare variants with more discrete distinction between allele calls and sensitivity for low-frequency (high-intensity) loci. All data were recalled for HG19 using GenomeStudio® v2011.1 (Illumina Inc., San Diego, CA, USA) per microarray platform and PLINK[78]. We note that these exome-based genotyping arrays consist of previously validated, high confidence rare variants, reducing the likelihood that findings might be the result of artifacts or genotyping errors that might affect sequencing studies. Further, since the imputation of very rare variants is difficult, only genotyped rare variants were used in this study; there were no imputed variants.

**Combination of cohorts for mega-analysis.** To increase power on rare variants, we sought to combine as many cohorts as possible into a mega-analysis. We thus performed principal components analysis (PCA) on all our cohorts after pruning the datasets for linkage disequilibrium using the pcair, part of the R package GENESIS. Pcair is designed to perform PCA in samples with cryptic relatedness and provides accurate ancestry inference that is not confounded by family structure[79]. For reference, we included individuals from all 11 HapMap reference panels in the PCA.

PCA showed two major groupings based on known ethnicity. The first consisted of the Han Chinese SCES and Malaysian SiMES cohorts, which were combined into the Eastern Asian combined cohort (EACC); we realize that technically the Malaysian population are Southeast Asians, but for simplicity will refer to this cohort as Eastern Asian. The second dataset consisted of the eight European cohorts (RSI, Croatia-Korcula, FITSA, EGCUT, TwinsUK, ERF, AREDS, and Ogliastra) and the one Indian cohort (SINDI). These cohorts were combined into the Indo-European combined cohort (IECC).

Analysis was performed on five discrete cohorts – IECC, EACC, EPIC-Norfolk, BDES, and REHS. The IECC analysis was performed in the Netherlands, while the EACC was performed in the United States as well as in the Netherlands. The BDES, EPIC-Norfolk, and REHS analyses were performed in their countries of origin (the United States, the United Kingdom, and Australia, respectively) as was legally required; these studies served on a per study basis as replication cohorts. A breakdown of all cohorts and the combined cohort with which they are grouped is provided in Supplementary Data 1.

**Statistics and reproducibility.** Quality control of the genotype data was performed as follows. For the combined cohorts, all raw cohort data were merged into a single file. All five cohorts then underwent identical quality control using PLINK[78]. Any individual not genotyped at 99% of all variants was removed and any variant not genotyped at 99% was also removed. Variants with a HWE $p$-value less than a Bonferroni-corrected $p$-value (defined as 0.05/total number of variants in the dataset) were also excluded. We also checked for batch effects and calculated the identity-by-descent (IBD) value of all individuals in the cohort, removing duplicates and twins. Many of the datasets exhibited cryptic relatedness amongst subjects (especially the Ogliastra study, which enrolled participants on the Italian island of Sardinia). Related individuals were not removed from the cohorts, as our analysis methods corrected for relatedness.

After QC, IECC had 13,097 individuals with 150,619 variants, EACC had 4867 individuals with 98,750 variants, BDES had 1740 individuals with 105,671 variants, REHS had 1,020 individuals with 92,313 variants, and EPIC-NORFOLK had 6282 individuals with 637,160 variants.

The refractive error phenotype analyzed here was defined as the quantitative phenotype mean spherical equivalent (SER), measured in diopters (D). Refractive error measurements in both eyes were taken from all participants and SER was calculated by adding the spherical refractive error + half the cylindrical refractive error in each eye, then taking the mean of both eyes. Individuals who had undergone procedures that could alter refraction, e.g., cataract surgery, laser refractive error procedures, retinal detachment surgery, and other ophthalmic conditions that may influence refraction were excluded from these analyses. The average spherical equivalents and standard deviations of each cohort are provided in Supplementary Data 1.

Gene-based association analysis was performed using a gene-based version of EMMAX[80,81]. EMMAX uses a kinship matrix to correct for population stratification and cryptic relatedness, which are present in these cohorts. EMMAX has been modified to perform gene-based burden-style tests, including the variable threshold (VT)[29] and the combined multivariate and collapsing (CMC)[28] methods through the software EPACTS (https://genome.sph.umich.edu/wiki/EPACTS), which we will term EMMAX-VT and EMMAX-CMC, respectively[82].

We analyzed all five cohorts with EMMAX-VT and EMMAX-CMC using genetic variants with a maximum MAF = 0.01. We only included variants that were in an exon of a gene (as defined by RefSeq), including both nonsynonymous and synonymous variants. Common variants (MAF > 0.01) and variants with a MAF ≤ 0.01 that mapped to an intergenic region were excluded from the analysis. Any gene with a minor allele count (MAC) of less than three for the cohort was dropped from the analysis.

Initial analyses were performed without any covariates. We performed two follow-up analyses using age, sex, and education level (low, intermediate, and high). One covariate analysis included all three covariates, while the second used age and sex only (education level removed). We note that the inclusion of covariates resulted in no significant difference between significant genes; for brevity we only

discuss the results without covariates. In addition, the Ogliastra cohort did not have data on age and education, thus ~3000 individuals were removed from the IECC covariate analyses. Hence, the covariate analyses are underpowered with respect to the analyses without covariates

We also performed gene-based analysis using the Aggregated Cauchy Association Test (ACAT)[77]. ACAT is a novel method that allows individual $p$-values to be combined into a gene-based $p$-value that is particularly useful for rare variants. To take advantage of this method, we analyzed all variants with a MAF ≤ 0.01 (with a minimum allele count of 3) using the original, single variant-based version of EMMAX[80,81]. We then combined the EMMAX $p$-values for each gene using the ACAT package implemented through R. Only nonsynonymous and synonymous exonic variants were included in the analysis.

Meta-analysis was then performed across our discovery cohorts. The burden-style tests that created a single $p$-value for a gene precluded the use of popular meta-analysis programs such as METAL, which require the input of reference and alternative alleles. Instead the gene-based $p$-values from the EMMAX-VT, EMMAX-CMC and ACAT were combined across studies using the classic method described by Fisher[83]. Fisher's method was implemented through the R package metap[84]. We defined genome-wide significant as $1 \times 10^{-5}$, based on the standard for gene-based studies. Replication was defined as a gene having a $P \le 0.05$ in one cohort after being found to be genome-wide significant in one of the other four cohorts. We note that this replication value is liberal and may lead to an inflation in false positives. However, as this is a discovery analysis, we were willing to allow some extra false positives in order to capture as many true positives as possible. A more stringent replication $p$-value of 3.9e−04 was also used to adjust for 129 attempted replications and these more stringently replicated genes were also reported.

We performed two separate meta-analyses. The first combined all five cohorts (IECC, EACC, BDES, EPIC-Norfolk, and REHS), which will be referred to as the multiethnic meta-analysis. The second combined the four ethnically Indo-European cohorts (IECC, BDES, REHS, and EPIC-Norfolk), which will be referred to as the Indo-European meta-analysis. The Indo-European meta-analysis was designed to identify any genes that might be significant in Indo-European-derived individuals but not significant in Eastern Asians; thus, we also report the Eastern Asian analyses $p$-values.

To investigate whether signals identified by the rare variant analysis were being partially driven by common variants, we calculated polygenic risk scores (PRS) for all cohorts using common variants identified in previous GWAS[22]. PRS were calculated for each subject using PLINK (Supplementary Data 2). All rare variant analyses were then repeated using the PRS values for each subject as a covariate. We compared the explained variance ($R^2$) of our top individual genes between the analysis with and without including PRS (Supplementary Data 3 and 4).

Independent replication of the genome-wide significant genes was performed in the UK Biobank (UKBB) via extraction of all rare variants comprising the genome-wide significant genes and repeating the same analyses.

**Pathway and expression analysis.** All genome-wide significant genes in the four meta-analyses and the EACC analyses were analyzed using Ingenuity Pathway Analysis (IPA) (QIAGEN Inc., https://digitalinsights.qiagen.com/products-overview/discovery-insights-portfolio/analysis-and-visualization/qiagen-ipa/)[85]. We performed various analyses through IPA, including canonical pathway analysis (identifying which genes are in known pathways), upstream regulator analysis (which identifies genes, RNAs, and proteins that regulate the genes in the dataset), and causal network analysis (which expands the pathway analysis to include the upstream regulators in the pathway analysis). IPA also identified disease phenotypes, cellular/molecular functions, and physiological networks associated with the genes in the dataset. Additional pathway and expression analysis were also performed with Functional Mapping and Annotation of GWAS[86,87] (FUMA), which provided tissue-enrichment information from GTEx and gene-group information from MsigDB. We repeated the IPA and FUMA analyses for our top prioritized genes from the schema proposed below.

**Gene prioritization based on biological function.** To prioritize genes according to biological background, we evaluated genes following a modified schedule proposed by Fritsche et al.[88] and further adapted by Tedja et al.[21]. Genes were ranked based on points equally assigned for the presence of replication, expression and biological plausibility. Evidence for ocular expression was based on single-cell expression data from adult human retina and developed organoids[47]. Biological plausibility was based on the presence of an ocular phenotype in OMIM and/or DisGeNET[89] as well as an ocular phenotype in a knock-out mouse model of this gene (Mouse Genome Informatics and International Mouse Phenotyping Consortium databases). The prioritization score ranged from zero to seven. In addition, we performed a look-up of the top-genes to screen for drugs that had these genes as target using SuperTarget[90], PharmGkb[91], STITCH v5.0[92] and DrugBank v5.0[93].

**Variant annotation for potential causal variants.** We performed annotation to identify potential causal variants within the significant genes. Therefore, we annotated all exonic variants from genome-wide significant genes using wANNOVAR[94–96], which collates functional predictions from popular prediction

algorithms like SIFT[97], PolyPhen2[98], MutationTaster[99], CADD[100], and FATHMM[101]. We initially looked at the top-ranked genes in the prioritization approach described above, giving preference to variants that appeared to either be driving the gene-based association analysis or variants that the five annotation algorithms agreed upon as being damaging. We further expanded this approach to all significant genes identified in the meta-analyses.

**Structural analysis of variants**. We also performed structural analysis of all coding variants within our top prioritized genes, as well as all mutations predicted to be deleterious in all genome-wide significant genes. We examined 1) all coding variants tested within six prioritized novel candidate myopia genes and 2) the predicted deleterious variants in *USH2A*, a non-prioritized but genome-wide significant gene with the highest number of predicted deleterious variants. The first set comprises 98 variants across 6 proteins, including 26 of special interest, which were looked at more closely and are covered below. Those are labeled in yellow in figures and represent missense variants predicted to have a deleterious effect by at least one commonly used variant effect predictor-tool (Table 2) or/and which displayed single variant association *p*-value below the nominal threshold of 5%. Crystal structures were obtained from the Protein Data Bank;[102] when crystal structures were not available, homology models were used for visualization and energy calculations. We used both FoldX RepairPDB and Position Scan[103] to predict differences in free energy between the wildtype and mutant proteins ($\Delta\Delta G$, measured in kcal/mol). ChimeraX[104] was used to visualize affected proteins. We also incorporated prior information from publicly available databases (OMIM, PFam, ClinVar, gnomAD, UniProt, RCSB PDB) and predicted functional effects (Missense3D[105]). A more detailed explanation of each individual protein can be found in Supplementary Note 2.

**Reporting summary**. Further information on research design is available in the Nature Portfolio Reporting Summary linked to this article.

## Data availability

The data that support the findings of this study are not publicly available due to information that could compromise research participant privacy and/or consent. European Union data privacy rulings currently forbid sharing of genomic data outside the EU and several of the participating studies have additional restrictions to protect the privacy of the study participants. Deidentified data were used here under data use agreements with each participating study. Data may be available by request from the individual participating studies if all regulatory conditions are met. Summary level data is provided in the Supplementary Data (Supplementary Data 6–13). Summary statistics for the multi-ethnic meta-analysis have been deposited in the GWAS catalog (https://www.ebi.ac.uk/gwas/downloads) with accession number GCST90244057.

## Code availability

The R scripts used to perform the analyses in this study are available in Supplementary Software 1.

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

## Acknowledgements

The authors gratefully acknowledge Sana Wajid of the Bioinformatics Core of the University of Pennsylvania for her quality control work on these data. This work was funded in part by the Intramural Research Program of the National Human Genome Research Institute, National Institutes of Health. The acknowledgments for each individual study cohort are given alphabetically by study below. A.P.K. is supported by a UKRI Future Leaders Fellowship. Molecular graphics and analyses were performed with UCSF ChimeraX, developed by the Resource for Biocomputing, Visualization, and Informatics at the University of California, San Francisco, with support from National Institutes of Health R01-GM129325 and the Office of Cyber Infrastructure and Computational Biology, National Institute of Allergy and Infectious Diseases. *AREDS*: AREDS was supported by the National Eye Institute (grants R01EY16482, R21EY015145, and P30EY11373) and by Research to Prevent Blindness and the Ohio Lions Eye Research Foundation. AREDS was also supported by contracts from National Eye Institute/National Institutes of Health, Bethesda, MD, with additional support from Bausch & Lomb Inc, Rochester, NY. The genotyping costs were supported by the National Eye Institute (R01EY020483 to D.S.) and some of the analyses were supported by the Intramural Research Program of the National Human Genome Research Institute, National Institutes of Health, USA. AREDS acknowledges Frederick Ferris, National Eye Institute, National Institutes of Health, Bethesda, MD; and the Center for Inherited Disease Research, Baltimore, MD where SNP genotyping was carried out. The investigators gratefully acknowledge the advice and guidance of Hemin Chin of the National Eye Institute. *BDES*: BDES was supported by the National Eye Institute of the National Institutes of Health under award numbers EY06594 (R. Klein and B. E. K. Klein), EY10605 (B. E. K. Klein) and R01EY021531 (A.P.K. and P.D.) and some of the analyses were supported by the Intramural Research Program of the National Human Genome Research Institute, National Institutes of Health, USA. *Croatia-Korcula*: The Croatia-Korcula study was funded by the Medical Research Council (UK) "QTL in health and disease" programme core grants, currently MC_UU_00007/10, as well as grants from the Republic of Croatia Ministry of Science, Education and Sports (108-1080315-0302; 216-1080315-0302) and the Croatian Science Foundation (8875). The study acknowledge Dr. Biljana Andrijević Derk, Valentina Lacmanović Lončar, Krešimir Mandić, Antonija Mandić, Ivan Škegro, Jasna Pavičić Astaloš, Ivana Merc, Miljenka Martinović, Petra Kralj, Tamara Knežević and Katja Barać-Juretić as well as the recruitment team from the Croatian Centre for Global Health, University of Split and the Institute of Anthropological Research in Zagreb for the ophthalmological data collection; the Wellcome Trust Clinical facility (Edinburgh, United Kingdom) for Exome array genotyping. *EGCUT*: EGCUT was supported by the European Union H2020 grant 692145, Est.RC grant IUT20-60 and the European Regional Development Fund, in the frame of Centre of Excellence in Genomics and Estonian Research Infrastructure's Roadmap and the University of Tartu (SP1GVARENG). This research was supported by NIH grant 5R01 DK07 57 87 -13, under subward-agreement GENFDOOO1B52751; the European Union through Horizon 2020 research and innovation programme under grant 633589 and the European Regional Development Fund (Project No. 2014-2020.4.01.16-0125). This research was also supported by the European Union through the European Regional Development Fund (Project No. 2014-2020.4.01.16-0125) and the Estonian Research Council grant PUT (PRG687) European Union H2020 grant 654248 (Corbel). EGCUT acknowledges the High Performance Computing Center of the University of Tartu. *EPIC-Norfolk*: The EPIC-Norfolk study (https://doi.org/10.22025/2019.10.105.00004) has received funding from the Medical Research Council (MR/N003284/1 and MC-UU_12015/1) and Cancer Research UK (C864/A14136). The genetics work in the EPIC-Norfolk study was funded by the Medical Research Council (MC_PC_13048).We are grateful to all the participants who have been part of the project and to the many members of the study teams at the University of Cambridge who have enabled this research.

*FITSA*: FITSA was supported by ENGAGE (FP7-HEALTH-F4-2007, 201413); European Union through the GENOMEUTWIN project (QLG2-CT-2002-01254); the Academy of Finland Center of Excellence in Complex Disease Genetics (213506, 129680); the Academy of Finland Ageing Programme; and the Finnish Ministry of Culture and Education and University of Jyväskylä. FITSA acknowledges the contributions of Emmi Tikkanen, Samuli Ripatti, Markku Kauppinen, Taina Rantanen and Jaakko Kaprio.

*Ogliastra*: The Ogliastra Study gratefully acknowledges the population of Ogliastra, Sardinia, Italy. The Ogliastra study was funded by a grant from the Italian Ministry of Education, University and Research (MIUR) n°: 5571/DSPAR/2002.

*RSI, ERF*: The Rotterdam Study and ERF were supported by European Research Council (ERC) under the European Union's Horizon 2020 research and innovation programme (grant 648268), Netherlands Organisation for Scientific Research (NWO, grant 91815655 to C.C.W.K. and NWO Veni 91617076 to V.J.M.V.), Ammodo Award (to C.C.W.K.), Erasmus Medical Center and Erasmus University, Rotterdam, The Netherlands; Netherlands Organization for Health Research and Development (ZonMw); the Research Institute for Diseases in the Elderly; the Ministry of Education, Culture and Science; the Ministry for Health, Welfare and Sports; the European Commission (DG XII); the Municipality of Rotterdam; the Netherlands Genomics Initiative/NWO; Center for Medical Systems Biology of NGI; Jacoba Breen Fonds, Topcon Europe; Ada Hooghart, Corina Brussee, Riet Bernaerts-Biskop, Amal Hamimida, Patricia van Hilten, Pascal Arp, Jeanette Vergeer, Sander Bervoets. The generation and management of the Illumina exome chip v1.0 array data for the Rotterdam Study (RS-I) was executed by the Human Genotyping Facility of the Genetic Laboratory of the Department of Internal Medicine, Erasmus MC, Rotterdam, The Netherlands. The Exome chip array data set was funded by the Genetic Laboratory of the Department of Internal Medicine, Erasmus MC, from the Netherlands Genomics Initiative (NGI)/Netherlands Organisation for Scientific Research (NWO)-sponsored Netherlands Consortium for Healthy Aging (NCHA; project nr. 050-060-810); the Netherlands Organization for Scientific Research (NWO; project number 184021007) and by the Rainbow Project (RP10; Netherlands Exome Chip Project) of the Biobanking and Biomolecular Research Infrastructure Netherlands (BBMRI-NL; www.bbmri.nl). We thank Ms. Mila Jhamai, Ms. Sarah Higgins, and Mr. Marijn Verkerk for their help in creating the exome chip database. The authors are grateful to the study participants, the staff from the Rotterdam Study and the participating general practitioners and pharmacists.

*REHS*: The core management of the Raine Study is funded by the University of Western Australia, Australia; the Telethon Institute for Child Health Research, Australia; Raine Medical Research Foundation, Australia; Women's and Infant's Research Foundation, Australia; Curtin University, Australia; Murdoch University, Australia; Edith Cowan University, Australia; and the University of Notre Dame, Australia. The Generation-2 20-year follow-up of the Raine Study was funded by the National Health and Medical Research Council (NHMRC), Australia: project grant no.: 1 021 105. The Generation-2 28-year follow-up of the Raine Study was funded by the NHMRC, Australia: project grants 1 121 979 and 1 126 494.

*SCES, SiMES, SINDI*: The Singapore studies (SCES, SiMES, SINDI) were supported by the National Medical Research Council, Singapore (NMRC 0796/2003, NMRC 1176/2008, STaR/0003/2008; CG/SERI/2010), Biomedical Research Council, Singapore (06/1/21/19/466, 09/1/35/19/616 and 08/1/35/19/550). The Singapore Tissue Network and the Genome Institute of Singapore, Agency for Science, Technology and Research, Singapore provided services.

*TwinsUK*: TwinsUK received funding from the Wellcome Trust; the European Union MyEuropia Marie Curie Research Training Network; Guide Dogs for the Blind Association; the European 18 Community's FP7 (HEALTHF22008201865GEFOS); ENGAGE (HEALTHF42007201413); the FP-5 GenomEUtwin Project (QLG2CT200201254); US National Institutes of Health/National Eye Institute (1RO1EY018246); NIH Center for Inherited Disease Research; the National Institute for Health Research comprehensive Biomedical Research Centre award to Guy's and St. Thomas' National Health Service Foundation Trust partnering with King's College London. P.G.H. is the recipient of a Fight for Sight ECI award. We acknowledge the contribution of Drs Toby Andrew, Margarida Lopes, Samantha Fahy and Diana Kozareva.

## Author contributions

A.M.M.: designed the study, performed analyses, wrote the manuscript. A.E.G.H.: designed the study, performed analyses, wrote the manuscript. R.N.L.: performed analyses on EPIC-Norfolk cohort. J.-S.O.: performed analyses on REHS cohort. K.P.: performed analyses on the UK BioBank. R.H.T.: performed analyses. J.M.: performed analyses. I.J.: performed analyses. R.J.: performed analyses. P.Z.W.: performed analyses. D.D.L.: designed the study, performed analyses, edited the manuscript. M.S.T.: performed analyses. A.I.I.: performed analyses. H.L.: performed analyses. C.S.C.: provided expression data. CREAM authors: designed the study, provided data, edited the manuscript. G.B.: provided data, designed the study, edited the manuscript. A.P.K.: provided data, designed the study, edited the manuscript. P.D.: provided data, designed the study, edited the manuscript. D.A.M.: provided data, designed the study, edited the manuscript. C.H.: provided data, designed the study, edited the manuscript. T.H.: provided data, designed the study, edited the manuscript. A.M.: provided data, designed the study,

edited the manuscript. J.W.: provided data, designed the study, edited the manuscript. O.P.: provided data, designed the study, edited the manuscript. C.-Y.C.: provided data, designed the study, edited the manuscript. S.-M.S.: provided data, designed the study, edited the manuscript. D.S.: provided data, designed the study, edited the manuscript. P.G.H.: provided data, designed the study, performed analyses, edited the manuscript. A.P.K.: provided data, designed the study, edited the manuscript. V.V.: provided data, designed the study, edited the manuscript. C.J.H.: provided data, designed the study, edited the manuscript. C.M.vD.: designed the study, edited the manuscript. V.J.M.V.: provided data, designed the study, edited the manuscript. C.C.W.K.: provided data, designed the study, edited the manuscript. J.E.B.W.: provided data, designed the study, edited the manuscript.

## Funding

## Competing interests
The authors declare no competing interests.

## Additional information

[1]Computational and Statistical Genomics Branch, National Human Genome Research Institute, National Institutes of Health, Baltimore, MD, USA. [2]Department of Ophthalmology, Erasmus Medical Center, Rotterdam, The Netherlands. [3]Department of Epidemiology, Erasmus Medical Center, Rotterdam, The Netherlands. [4]MRC Epidemiology, University of Cambridge School of Clinical Medicine, Cambridge, UK. [5]NIHR Biomedical Research Centre, Moorfields Eye Hospital NHS Foundation Trust and UCL Institute of Ophthalmology, London, UK. [6]Statistical Genetics Laboratory, Department of Genetics and Computational Biology, QIMR Berghofer Medical Research Institute, Brisbane, QLD, Australia. [7]Department of Twin Research and Genetic Epidemiology, King's College London, London, UK. [8]MRC Human Genetics Unit, Institute of Genetics and Cancer, University of Edinburgh, Western General Hospital, Edinburgh, UK. [9]Institute for Biomedical Sciences, Perelman School of Medicine, University of Pennsylvania, Philadelphia, PA, USA. [10]Data Science Unit, Singapore Eye Research Institute, Singapore National Eye Centre, Singapore, Singapore. [11]Institute for Molecular and Clinical Ophthalmology Basel, Basel, Switzerland. [12]Institute of Molecular Genetics, National Research Council of Italy, Pavia, Italy. [13]Department of Epidemiology, Johns Hopkins University Bloomberg School of Public Health, Baltimore, MD, USA. [14]The Bloomberg School of Public Health, Johns Hopkins University, Baltimore, MD, USA. [15]Centre for Ophthalmology and Visual Science, Lions Eye Institute, University of Western Australia, Perth, WA, Australia. [16]Estonian Genome Center, Institute of Genomics, University of Tartu, Tartu, Estonia. [17]Department of Ophthalmology, University of Helsinki and Helsinki University Hospital, Helsinki, Finland. [18]Department of Public Health, University of Helsinki, Helsinki, Finland. [19]Department of Ophthalmology, Central Hospital of Central Finland, Jyväskylä, Finland. [20]Gerontology Research Center, Faculty of Sport and Health Sciences, University of Jyväskylä, Jyväskylä, Finland. [21]Centre for Quantitative Medicine, DUKE-National University of Singapore, Singapore, Singapore. [22]Ocular Epidemiology Research Group, Singapore Eye Research Institute, Singapore National Eye Centre, Singapore, Singapore. [23]Saw Swee Hock School of Public Health, National University Health Systems, National University of Singapore, Singapore, Singapore. [24]Myopia Research Group, Singapore Eye Research Institute, Singapore National Eye Centre, Singapore, Singapore. [25]Department of Ophthalmology, University of Pennsylvania, Philadelphia, PA, USA. [26]Nuffield Department of Population Health, University of Oxford, Oxford, UK. [27]Department of Clinical Genetics, Erasmus Medical Center, Rotterdam, The Netherlands. [28]Department of Ophthalmology, Radboud University Medical Centre, Nijmegen, The Netherlands. [81]These authors contributed equally: Anthony M. Musolf, Annechien E. G. Haarman. [82]These authors jointly supervised this work: Cornelia M. van Duijn, Virginie J. M. Verhoeven, Caroline C. W. Klaver, Joan E. Bailey-Wilson. *A list of authors and their affiliations appears at the end of the paper. ✉email: v.verhoeven@erasmusmc.nl; c.c.w.klaver@erasmusmc.nl; jebw@mail.nih.gov

## Consortium for Refractive Error and Myopia (CREAM)

Joan E. Bailey-Wilson [1]✉, Paul Nigel Baird[29], Amutha Barathi Veluchamy[30,31,32], Ginevra Biino[12], Kathryn P. Burdon[33], Harry Campbell[34], Li Jia Chen[35], Ching-Yu Cheng[21,22,36], Emily Y. Chew[37], Jamie E. Craig[38], Phillippa M. Cumberland[39], Margaret M. Deangelis[40], Cécile Delcourt[41], Xiaohu Ding[42], Priya Duggal [14], Cornelia M. van Duijn[26], David M. Evans[43,44,45], Qiao Fan[46], Maurizio Fossarello[47], Paul J. Foster[5], Puya Gharahkhani[48], Adriana I. Iglesias[2,3,27], Jeremy A. Guggenheim[49], Xiaobo Guo[42,50], Annechien E. G. Haarman [2,3], Toomas Haller [16], Christopher J. Hammond [7], Xikun Han[48], Caroline Hayward [8], Mingguang He[29,42], Alex W. Hewitt[15,29,33], Quan V. Hoang[30,51], Pirro G. Hysi [7], Sudha K. Iyengar[52,53,54], Jost B. Jonas[55,56], Mika Kähönen[57,58],

Jaakko Kaprio[17,59], Anthony P. Khawaja [4,5], Caroline C. W. Klaver [2,3,11,28✉], Alison P. Klein [13], Barbara E. Klein[60], Jonathan H. Lass[52,53], Kris Lee[60], Terho Lehtimäki[61,62], Deyana Lewis[1], Qing Li[1], Shi-Ming Li[56], Leo-Pekka Lyytikäinen[61,62], Stuart MacGregor[48], David A. Mackey[15,29,33], Nicholas G. Martin[63], Akira Meguro[64], Andres Metspalu [16], Candace Middlebrooks[1], Masahiro Miyake[65], Nobuhisa Mizuki[64], Anthony Musolf[1], Stefan Nickels[66], Konrad Oexle[67], Chi Pui Pang[35], Olavi Pärssinen[19,20], Andrew D. Paterson[68], Craig Pennell[15], Norbert Pfeiffer[66], Ozren Polasek[69,70], Jugnoo S. Rahi[5,39,71], Olli Raitakari[72,73], Igor Rudan[34], Srujana Sahebjada[29], Seang-Mei Saw[23,24], Dwight Stambolian[25], Claire L. Simpson[74], E-Shyong Tai[24], Milly S. Tedja[2,3], J. Willem L. Tideman[2,3], Akitaka Tsujikawa[65], Virginie J. M. Verhoeven [2,3,27✉], Veronique Vitart [8], Ningli Wang[56], Juho Wedenoja[17,18], Wen Bin Wei[75], Cathy Williams[45], Katie M. Williams[7], James F. Wilson[8,34], Robert Wojciechowski[14,76], Ya Xing Wang[56], Kenji Yamashiro[77], Jason C. S. Yam[35], Maurice K. H. Yap[78], Seyhan Yazar[15], Shea Ping Yip[79], Terri L. Young[60] & Xiangtian Zhou[80]

[29]Centre for Eye Research Australia, Ophthalmology, Department of Surgery, University of Melbourne, Royal Victorian Eye and Ear Hospital, Melbourne, VIC, Australia. [30]Singapore Eye Research Institute, Singapore National Eye Centre, Singapore, Singapore. [31]Duke-NUS Medical School, Singapore, Singapore. [32]Department of Ophthalmology, National University Health Systems, National University of Singapore, Singapore, Singapore. [33]Department of Ophthalmology, Menzies Institute of Medical Research, University of Tasmania, Hobart, TAS, Australia. [34]Centre for Global Health Research, Usher Institute for Population Health Sciences and Informatics, University of Edinburgh, Edinburgh, UK. [35]Department of Ophthalmology and Visual Sciences, The Chinese University of Hong Kong, Hong Kong Eye Hospital, Kowloon, Hong Kong. [36]Department of Ophthalmology, Yong Loo Lin School of Medicine, National University of Singapore, Singapore, Singapore. [37]Division of Epidemiology and Clinical Applications, National Eye Institute/National Institutes of Health, Bethesda, MD, USA. [38]Department of Ophthalmology, Flinders University, Adelaide, SA, Australia. [39]Great Ormond Street Institute of Child Health, University College London, London, UK. [40]Department of Ophthalmology, University at Buffalo, Buffalo, NY, USA. [41]Université de Bordeaux, Inserm, Bordeaux Population Health Research Center, team LEHA, UMR 1219, F-33000 Bordeaux, France. [42]State Key Laboratory of Ophthalmology, Zhongshan Ophthalmic Center, Sun Yat-sen University, Guangzhou, China. [43]Translational Research Institute, University of Queensland Diamantina Institute, Brisbane, QLD, Australia. [44]MRC Integrative Epidemiology Unit, University of Bristol, Bristol, UK. [45]Department of Population Health Sciences, Bristol Medical School, Bristol, UK. [46]Centre for Quantitative Medicine, DUKE-National University of Singapore, Singapore, Singapore. [47]University Hospital 'San Giovanni di Dio', Cagliari, Italy. [48]Statistical Genetics, QIMR Berghofer Medical Research Institute, Brisbane, QLD, Australia. [49]School of Optometry & Vision Sciences, Cardiff University, Cardiff, UK. [50]Department of Statistical Science, School of Mathematics, Sun Yat-Sen University, Guangzhou, China. [51]Department of Ophthalmology, Columbia University, New York, NY, USA. [52]Department of Population and Quantitative Health Sciences, Case Western Reserve University, Cleveland, OH, USA. [53]Department of Ophthalmology and Visual Sciences, Case Western Reserve University and University Hospitals Eye Institute, Cleveland, OH, USA. [54]Department of Genetics, Case Western Reserve University, Cleveland, OH, USA. [55]Department of Ophthalmology, Medical Faculty Mannheim of the Ruprecht-Karls-University of Heidelberg, Mannheim, Germany. [56]Beijing Tongren Eye Center, Beijing Tongren Hospital, Beijing Institute of Ophthalmology, Beijing Key Laboratory of Ophthalmology and Visual Sciences, Capital Medical University, Beijing, China. [57]Department of Clinical Physiology, Tampere University Hospital and School of Medicine, University of Tampere, Tampere, Finland. [58]Finnish Cardiovascular Research Center, Faculty of Medicine and Life Sciences, University of Tampere, Tampere, Finland. [59]Institute for Molecular Medicine Finland FIMM, HiLIFE Unit, University of Helsinki, Helsinki, Finland. [60]Department of Ophthalmology and Visual Sciences, University of Wisconsin–Madison, Madison, WI, USA. [61]Department of Clinical Chemistry, Finnish Cardiovascular Research Center-Tampere, Faculty of Medicine and Life Sciences, University of Tampere, Tampere, Finland. [62]Department of Clinical Chemistry, Fimlab Laboratories, University of Tampere, Tampere, Finland. [63]Genetic Epidemiology, QIMR Berghofer Medical Research Institute, Brisbane, QLD, Australia. [64]Department of Ophthalmology, Yokohama City University School of Medicine, Yokohama, Kanagawa, Japan. [65]Department of Ophthalmology and Visual Sciences, Kyoto University Graduate School of Medicine, Kyoto, Japan. [66]Department of Ophthalmology, University Medical Center of the Johannes Gutenberg University Mainz, Mainz, Germany. [67]Institute of Neurogenomics, Helmholtz Zentrum München, German Research Centre for Environmental Health, Neuherberg, Germany. [68]Program in Genetics and Genome Biology, Hospital for Sick Children and University of Toronto, Toronto, ON, Canada. [69]Gen-info Ltd, Zagreb, Croatia. [70]University of Split School of Medicine, Soltanska 2, Split, Croatia. [71]Ulverscroft Vision Research Group, University College London, London, UK. [72]Research Centre of Applied and Preventive Cardiovascular Medicine, University of Turku, Turku, Finland. [73]Department of Clinical Physiology and Nuclear Medicine, Turku University Hospital, Turku, Finland. [74]Department of Genetics, Genomics and Informatics, University of Tennessee Health Sciences Center, Memphis, TN, USA. [75]Beijing Tongren Eye Center, Beijing Key Laboratory of Intraocular Tumor Diagnosis and Treatment, Beijing Ophthalmology & Visual Sciences Key Lab, Beijing Tongren Hospital, Capital Medical University, Beijing, China. [76]Wilmer Eye Institute, Johns Hopkins Medical Institutions, Baltimore, MD, USA. [77]Department of Ophthalmology, Otsu Red Cross Hospital, Nagara, Japan. [78]Centre for Myopia Research, School of Optometry, The Hong Kong Polytechnic University, Hong Kong, Hong Kong. [79]Department of Health Technology and Informatics, The Hong Kong Polytechnic University, Hong Kong, Hong Kong. [80]School of Ophthalmology and Optometry, Eye Hospital, Wenzhou Medical University, Wenzhou, China.

