## [Peer Review File · Communications Biology]

Reviewers' comments:

Reviewer #1 (Remarks to the Author):

1. In general, the prevalence of high myopia in East Asian populations is higher than in other populations. However, in this study, the East Asian combined cohort (EACC) has a less myopic mean SE than the Indo-European combined cohort (IECC). Are there any possible sampling biases in these cohorts?
2. Rare variants often have larger effects on gene function and as a result lead to large changes in disease risk or trait values. Were any of the identified 129 genes specifically associated with high myopia? The authors should add the results or at least comment on this.
3. The authors mentioned that "129 novel associated genes were identified" in the 1st paragraph of the Discussion, however that "Most of these genes are novel for association with refractive error" in the last paragraph of the Discussion. Please clarify whether or not all of the identified genes are novel for refractive error.
4. Considering the SE phenotypes of the study cohorts, this study may also identify rare variants for association with emmetropia or hyperopia. Were all the identified 129 genes associated with myopia? Or did the study identify variants associated with emmetropia or hyperopia? The authors should clarify this.
5. The authors mentioned that this study included only nonsynonymous and synonymous exonic variants in the analyses. They should clarify what kinds of variants from Exome Arrays were excluded.
6. Methods, Quality Control: Please show the HWE criteria in QC.
7. Supplementary Tables: The authors should show the cumulative minor allele count/frequency, and odds ratio or beta in the gene-based association tests.
8. Supplementary Tables 10 and 11: Their titles are the same "All multiethnic meta-analysis p-values for EMMAX-CMC". Supplementary Table 11 is probably for ACAT. Please correct.
9. Supplementary Tables 6-20: The results from the three gene-based analyses have not been presented in regular order. (Ex. Tables S12, S13, and S14 is for EMMAX-VT, ACAT, and EMMAX-CMC, respectively, whereas Tables S15, S16, and A17 is for EMMAX-VT, EMMAX-CMC, and ACAT.) This is very confusing. Please correct the order.

Reviewer #2 (Remarks to the Author):

Musolf et. al., conducted an exome sequencing study across 13 cohorts to find risk genes for ocular refractive error. Although genetic studies using SNP-based GWAS approaches have previously uncovered common risk-associated variants, they only explain 18% of observed variance. The impetus of this manuscript is therefore to use an exome-based sequencing approach to find rare genetic variants with large effect sizes to account for the remainder of the observed variance.

This appears to be the first large-scale rare variant analysis on refractive error using multiethnic cohorts. A total of 129 genes that were significantly linked to refractive error were found from analyzing exomes across cohorts and between cohorts of similar ethnicities by utilizing 3 different analytical methods. Although these data appears to be novel and despite the apparent rigor of this approach, there are four major concerns with this manuscript that limit its utility and interest to the field of refractive error research. These are outlined below:

- a) No Sanger sequencing was performed to verify rare allele calls. Given the large and disparate differences between the results of the three gene-based analysis method, and the varying replication rate of the candidate genes, it is more than plausible that the differences could be due to artifacts from analysis and/or noise due to sequencing error. Sanger sequencing must be performed to validate discovered variants, otherwise they are of limited utility for experimental follow-up.
- b) These authors claim that rare variants from these 129 gene candidates can account for the remainder of phenotypic variants observed. However, no effect size data is furnished. Without effect size data, this claim does not stand. Effect size data should also be used to prioritize key genes from

the pool of 129 gene candidates. High priority genes must not only be replicated in independent cohorts but must also fulfill a minimum effect size cut-off. There is no experimental value in studying a rare variant with an odds ratio of less than 1.5.

c) Given the large number of individuals and SER phenotype information available in the CREAM consortium dataset, the authors should also quantify the refractive error in individuals carrying risk gene variants as an orthogonal way to estimate effect size and prioritize risk genes. High priority genes with large effect sizes will be expected to strongly affect SER.

d) In this manuscript, the authors furnished expression data of risk genes mined from other databases. However, because no comparative expression studies of high priority genes were carried out on actual disease vs. control eyes, it is still unclear where risk genes are expressed in the eye, and whether they are altered in disease states. Unless the authors have a compelling reason to believe that genes expressed elsewhere could affect ocular phenotype, actual expression studies of RNA or protein should be carried out to ensure ocular relevance, instead of relying on mined data.

In the absence of these data, it is impossible to determine whether these 129 genes are bona fide risk genes for refractive error. Therefore, prioritization of these candidates using the above mentioned criteria, will add substantial value to this manuscript and enhance its utility to the research community.

Reviewer #3 (Remarks to the Author):

This manuscript focuses on exploring the role of rare variants in refractive error. GWAS has established the role of common variants as a risk factor for refractive index. However, the substantial heritability remains unexplained, which could be attributed to additional common variants, rare variants, or gene-gene interactions. This is a timely study to understand the contribution of the rare variants in refractive error. The authors have performed a multiethnic meta-analysis that identified 129 potential candidates. Replication in the other cohorts led to the identification of three new likely candidates. The manuscript is written well. I have a few comments that I would like the authors to address.

1. It is not clear whether all the rare variants included in the study were directly genotyped as part of the exome array or were imputed.

2. Authors state in the method section that "We only included variants that were in an exon of a gene (as defined by RefSeq), including both nonsynonymous and synonymous variants". It is not clear why the synonymous variants were used for the burden test as they are presumably largely benign and neutral and might not have any impact on gene function. In fact, they are often used to calibrate gene-based approaches to guard against type I error. Could authors comment on retaining synonymous variants in the burden analysis?

3. In the initial analysis, genome-wide significance was set as 1×10^{-5} , and the replication was defined as having a $P \leq 0.05$ in one cohort after being found to be genome-wide significant in one of the other four cohorts. However, this could lead to many false positives. In the replication at least, the correction should be applied for 129 genes that were discovered in the Indo-European meta-analysis.

4. The authors state that they did not observe any difference in the results upon inclusion of covariates and they decided to discuss the results without covariates. The inclusion of covariates in such analysis is a common practice and it might be better to present the results with those. Is there any particular reason to present the results without covariates?

5. One of the major limitations of the study is that the rare variants were limited to the ones that were part of the exome arrays as the authors point out in the discussion. Thus, the cohort is likely to harbor many more rare variants that were missed in the analysis.

6. For the expression data, the authors have limited their investigation to the retina. However, the refractive index can be caused by problems in the shape of the cornea or aging of the lens. Thus, the expression of the candidates in these tissues should also be considered for prioritization. Could authors comment on using only retina data here?

Minor comments:

1. The inclusion of the MAF in table 1 can be informative.

2. EPACTS toolbox includes VT, CMC and SKAT. However, the authors here only try the first two. Is there any reason for not using SKAT?

Resubmission - Rare variant analyses across multiethnic cohorts identify novel genes for refractive error.

Musolf et al., Communications Biology

We thank the reviewers for their helpful comments. We respond to each comment below.

Reviewer #1 (Remarks to the Author):

Comment 1: In general, the prevalence of high myopia in East Asian populations is higher than in other populations. However, in this study, the East Asian combined cohort (EACC) has a less myopic mean SE than the Indo-European combined cohort (IECC). Are there any possible sampling biases in these cohorts?

Response: This is a very good point by the reviewer, as East Asians tend to have a lower SER than other populations. There was no inherent bias in the sampling of the SCES and SiMES cohorts, both were a random sampling of Singaporean Chinese and Malay populations. It is possible that there is some bias due to age when compared to the IECC, as the SiMES and SCES studies collected individuals above 40, whereas many of the cohorts in IECC allowed individuals 18 and older. We also note that the EACC average SER was inflated primarily due to the Malay population, which had an average SER of -0.11.

Comment 2: Rare variants often have larger effects on gene function and as a result lead to large changes in disease risk or trait values. Were any of the identified 129 genes specifically associated with high myopia? The authors should add the results or at least comment on this.

Response: This is a very good point by the reviewer. Two of the genes identified by this study, GDF15 and USH2A, have been found to be associated with high myopia. This is now stated in the discussion in lines 552-555.

Comment 3: The authors mentioned that “129 novel associated genes were identified” in the 1st paragraph of the Discussion, however that “Most of these genes are novel for association with refractive error” in the last paragraph of the Discussion. Please clarify whether or not all of the identified genes are novel for refractive error.

Response: We regret any confusion from this statement. We have clarified the first sentence of the Discussion, which now reads ten genes were novel with ten being previously associated with refractive error, six being associated with myopia, and none with hyperopia. This is now stated in lines 551-555.

Comment 4: Considering the SE phenotypes of the study cohorts, this study may also identify rare variants for association with emmetropia or hyperopia. Were all the identified 129 genes associated with myopia? Or did the study identify variants associated with emmetropia or hyperopia? The authors should clarify this.

Response: This is a good point by the reviewer. As this was a quantitative analysis of refractive error, indeed variants might be associated with either myopia (negative SE) or hyperopia (positive refractive error). We have included the direction of effects for all genes for the CMC and VT tests in Supplementary Tables 8-9; ACAT does not test for directionality. We also added Supplementary table 27 which displays the mean SE for carriers versus non-carriers of the rare variants in our prioritized genes, which gives an indication of the effects of these rare variants on the mean SE trait with some variants showing myopic mean Se and some hyperopic mean SE. We also note in the Discussion, lines 661-663 that we plan further analyses where we convert our quantitative RE phenotype to binary phenotypes and test for association with myopia, hyperopia, and astigmatism.

Comment 5: The authors mentioned that this study included only nonsynonymous and synonymous exonic variants in the analyses. They should clarify what kinds of variants from Exome Arrays were excluded.

Response: This is a very good point by the reviewer. We have clarified that the variants that were excluded from the analysis were any common variants ($MAF > 0.01$) in any region of the genome or any rare variant ($MAF \leq 0.01$) that was located within an intergenic region. For the burden style tests, (VT and CMC) gene-based markers with a minor allele count less than 3 were excluded. For the single variant tests, variants with a minor allele count less than 3 were excluded. This is now stated in lines 215-218.

Comment 6: Methods, Quality Control: Please show the HWE criteria in QC.

Response: We thank the reviewer for this comment. We removed any variants that had a HWE p-value less than the Bonferroni corrected p-value (defined here as $0.05/\text{total number of variants in the dataset}$). This is now stated clearly in lines 186-188.

Comment 7: Supplementary Tables: The authors should show the cumulative minor allele count/frequency, and odds ratio or beta in the gene-based association tests.

Response: We agree with the reviewer on this point. For the CMC analysis, we have added the frequencies as well as the beta values for each gene. For the VT analysis, we have included the frequencies and directionality (+/-). VT does not output a beta value, because although it is burden test like CMC, VT uses multiple MAF thresholds, calculating a z score from each regression and taking the maximum with p-values determined by permutation. Both Fisher's method and the ACAT method combine p-values without inputting a beta, so beta is not output. This is now stated in lines 324-325 and the frequencies, rare variant counts, betas or direction of effect are included in Supplemental tables 9-11 for all genes and all cohorts.

Comment 8: Supplementary Tables 10 and 11: Their titles are the same "All multiethnic meta-analysis p-values for EMMAX-CMC". Supplementary Table 11 is probably for ACAT. Please correct.

Response: We thank the reviewer for identifying this error. We have corrected the title to “All multiethnic meta-analysis p-values for ACAT” as correctly suggested by the reviewer.

Comment 9: Supplementary Tables 6-20: The results from the three gene-based analyses have not been presented in regular order. (Ex. Tables S12, S13, and S14 is for EMMAX-VT, ACAT, and EMMAX-CMC, respectively, whereas Tables S15, S16, and A17 is for EMMAX-VT, EMMAX-CMC, and ACAT.) This is very confusing. Please correct the order.

Response: We regret that the order of the supplementary tables was confusing, and we have reordered them in the following manner: VT, CMC, ACAT for all tables.

Responses to Reviewer #2

Musolf et. al., conducted an exome sequencing study across 13 cohorts to find risk genes for ocular refractive error. Although genetic studies using SNP-based GWAS approaches have previously uncovered common risk-associated variants, they only explain 18% of observed variance. The impetus of this manuscript is therefore to use an exome-based sequencing approach to find rare genetic variants with large effect sizes to account for the remainder of the observed variance.

This appears to be the first large-scale rare variant analysis on refractive error using multiethnic cohorts. A total of 129 genes that were significantly linked to refractive error were found from analyzing exomes across cohorts and between cohorts of similar ethnicities by utilizing 3 different analytical methods. Although these data appears to be novel and despite the apparent rigor of this approach, there are four major concerns with this manuscript that limit its utility and interest to the field of refractive error research. These are outlined below:

Comment 1: No Sanger sequencing was performed to verify rare allele calls. Given the large and disparate differences between the results of the three gene-based analysis method, and the varying replication rate of the candidate genes, it is more than plausible that the differences could be due to artifacts from analysis and/or noise due to sequencing error. Sanger sequencing must be performed to validate discovered variants, otherwise they are of limited utility for experimental follow-up.

Response: This point by the reviewer has caused us to further clarify the information about the exome chip genotyping performed here. We note that our dataset does not contain whole exome sequence (WES) data, rather it consists of genotype data from highly enriched commercial exome SNP arrays. These SNPs have all been previously validated as high performing rare variant loci, thus severely reducing the potential for these variants to be artifacts or errors, as would be the case in sequencing data. This is now clearly explained on lines 156-159.

Comment 2: These authors claim that rare variants from these 129 gene candidates can account for the remainder of phenotypic variants observed. However, no effect size data is furnished. Without effect size data, this claim does not stand. Effect size data should also be used to prioritize key genes

from the pool of 129 gene candidates. High priority genes must not only be replicated in independent cohorts but must also fulfill a minimum effect size cut-off. There is no experimental value in studying a rare variant with an odds ratio of less than 1.5.

*Response: This is a good point by the reviewer. We did not mean to imply that the significantly associated genes in this analysis account for the remainder of phenotypic variation observed in the Mean Spherical Equivalent measure of refractive error and have clarified this in the revised manuscript. We agree that effect sizes are very informative and important information in an association study. **For the CMC analysis, we have added the frequencies as well as the beta values for each gene. For the VT analysis, we have included the frequencies and directionality (+/-).** VT does not output a beta value, because although it is burden test like CMC, VT uses multiple MAF thresholds, calculating a z score from each regression and taking the maximum with p-values determined by permutation. Both Fisher's method and the ACAT method combine p-values without inputting a beta, so beta is not output. Thus, we note that while effect sizes are indeed important as the reviewer points out, we cannot use them as the sole means to prioritize our genes simply because two of our methods do not use them. This is now stated in lines 324-325.*

Comment 3: Given the large number of individuals and SER phenotype information available in the CREAM consortium dataset, the authors should also quantify the refractive error in individuals carrying risk gene variants as an orthogonal way to estimate effect size and prioritize risk genes. High priority genes with large effect sizes will be expected to strongly affect SER.

Response: This is an excellent point by the reviewer. We have created a new table (Supplementary Table 27) that shows the average mean SER for both minor allele carriers and non-carriers for all variants in the six prioritized genes (PDCD6IP, MAPT, CHST6, GRHL2, USH2A, P4HTM). This indicates the relative effect size for carriers of each rare variant compared to those with the homozygous major genotype. This is now clearly stated in lines 385-387. In Supplementary Table 27, it can be seen that the mean SERs of many of these rare variants in our prioritized genes are quite different in carriers versus non-carriers of the rare variants in question.

Comment 4: In this manuscript, the authors furnished expression data of risk genes mined from other databases. However, because no comparative expression studies of high priority genes were carried out on actual disease vs. control eyes, it is still unclear where risk genes are expressed in the eye, and whether they are altered in disease states. Unless the authors have a compelling reason to believe that genes expressed elsewhere could affect ocular phenotype, actual expression studies of RNA or protein should be carried out to ensure ocular relevance, instead of relying on mined data.

In the absence of these data, it is impossible to determine whether these 129 genes are bona fide risk genes for refractive error. Therefore, prioritization of these candidates using the above mentioned criteria, will add substantial value to this manuscript and enhance its utility to the research community.

Response: These are excellent points by the reviewer. While we believe that these studies are beyond the scope of this initial discovery analysis, these studies should be performed in the future. However, we

would like to stress that these comparative studies are incredibly complex. They would require tissue from children from different stages of myopization to compare to control children of similar ages. Currently, we are not aware of any researchers that have obtained this type of tissue; they are not readily available.

Another issue is that such experiments would not just involve a simple comparison of myopic versus normal eyes; we do not know if expression of myopia genes change during the process of myopization and does that expression then stabilize upon reaching adulthood. There are also environmental features to consider, such as when the tissue was harvested, what was the stage of disease, etc.

Responses to Reviewer #3

This manuscript focuses on exploring the role of rare variants in refractive error. GWAS has established the role of common variants as a risk factor for refractive index. However, the substantial heritability remains unexplained, which could be attributed to additional common variants, rare variants, or gene-gene interactions. This is a timely study to understand the contribution of the rare variants in refractive error. The authors have performed a multiethnic meta-analysis that identified 129 potential candidates. Replication in the other cohorts led to the identification of three new likely candidates. The manuscript is written well. I have a few comments that I would like the authors to address.

Comment 1: It is not clear whether all the rare variants included in the study were directly genotyped as part of the exome array or were imputed.

Response: This is a good point by the reviewer. Since imputation for rare variants has been suggested to have higher rates of imputation error than for common variants, only directly genotyped variants were used in this analysis. This is now clearly stated in lines 159-160..

Comment 2: Authors state in the method section that “We only included variants that were in an exon of a gene (as defined by RefSeq), including both nonsynonymous and synonymous variants”. It is not clear why the synonymous variants were used for the burden test as they are presumably largely benign and neutral and might not have any impact on gene function. In fact, they are often used to calibrate gene-based approaches to guard against type I error. Could authors comment on retaining synonymous variants in the burden analysis?

Response: We thank the reviewer for this comment. We note that we performed analyses where we excluded synonymous variants and only included nonsynonymous variants as well as analyses where both types of variants were included. We decided to report the studies including the synonymous variants and nonsynonymous variants because more genes were recorded as significant. This suggested that some of these rare synonymous variants might have an effect on gene function or protein availability, perhaps by affecting protein folding due to differential t-RNA availability during translation (as some authors have suggested) or by some other mechanism. Since the aim of this study was

discovery, we wanted to include the greatest number of candidate genes possible for future replication attempts by other investigators.

Comment 3: In the initial analysis, genome-wide significance was set as 1×10^{-5} , and the replication was defined as having a $P \leq 0.05$ in one cohort after being found to be genome-wide significant in one of the other four cohorts. However, this could lead to many false positives. In the replication at least, the correction should be applied for 129 genes that were discovered in the Indo-European meta-analysis.

Response: This is certainly a very good point by the reviewer; this method could indeed lead to an inflation in false positives. If we leave the replication threshold to be the same as the significance threshold, then there are two instances of replication - MRPS27 in EPIC-Norfolk and REHS in the EMMAX-VT analysis and GDF15 in the IECC and REHS cohorts with the ACATS test. However, thresholds for replication are usually allowed to be lower than that of initial significance. As this was a discovery analysis, we were more willing to allow some potential false positives in order to capture as many true replications as possible. This is now clearly stated in lines 228-230. Using a more stringent replication p-value of 3.87×10^{-4} (correcting for 129 gene replication tests) yields replications of GDF15 for all three tests and MRPS27 (VT) with PDCD6IP (VT), MRPS27 (VT), NDC80 (VT) and LOXHD1 (ACAT) all having replication p-values very close to these thresholds (Supplemental Tables 9-11). This information has now been included in the paper in lines 241-245 in the methods section and lines 316-322 in the results section.

Comment 4: The authors state that they did not observe any difference in the results upon inclusion of covariates and they decided to discuss the results without covariates. The inclusion of covariates in such analysis is a common practice and it might be better to present the results with those. Is there any particular reason to present the results without covariates?

Response: The reviewer is certainly correct that including covariates is common practice. We present the results without covariates because the covariate analyses are underpowered with respect to the no covariate analyses. The Ogliastra cohort from the Italian island of Sardinia, did not have any covariate information (besides sex), thus the covariate analyses have a sample size of 3008 individuals less than the no covariate individuals. This is now clearly stated on lines 222-226.

Comment 5: One of the major limitations of the study is that the rare variants were limited to the ones that were part of the exome arrays as the authors point out in the discussion. Thus, the cohort is likely to harbor many more rare variants that were missed in the analysis.

Response: This is an excellent point by the reviewer. We mention this limitation in the Discussion, lines 631-633. We describe that there are almost certainly rare variants in these cohorts that were not genotyped in this study and that these ungenotyped variants not only may harbor additional risk variants but could account for lack of replication from other GWAS findings.

Comment 6: For the expression data, the authors have limited their investigation to the retina. However, the refractive index can be caused by problems in the shape of the cornea or aging of the lens. Thus, the expression of the candidates in these tissues should also be considered for prioritization. Could authors comment on using only retina data here?

Response: This is a good point by the reviewer. We searched for potential sources of expression data in sclera and cornea in humans but were not successful in obtaining sources. Thus, we were limited to only using retinal expression sources. We have added some sentences in the Discussion suggesting expression data from corneal and scleral tissue would be useful in the future for further prioritization of these candidate genes (lines 649-651).

Minor comments:

Minor Comment 1: The inclusion of the MAF in table 1 can be informative.

Response: We thank the reviewer for this comment. We have included MAF in Table 1.

Minor Comment 2: EPACTS toolbox includes VT, CMC and SKAT. However, the authors here only try the first two. Is there any reason for not using SKAT?

Response: We thank the reviewer for this comment. We were having technical issues regarding the EMMAX SKAT module of EPACTS and thus were unable to use it.

Reviewers' comments:

Reviewer #1 (Remarks to the Author):

The authors are congratulated to make significant improvement in this revised manuscript. The authors addressed most of concerns from the reviewers.

Reviewer #2 (Remarks to the Author):

Responses are in the uploaded review file

Comment 2: These authors claim that rare variants from these 129 gene candidates can account for the remainder of phenotypic variants observed. However, no effect size data is furnished. Without effect size data, this claim does not stand. Effect size data should also be used to prioritize key genes from the pool of 129 gene candidates. High priority genes must not only be replicated in independent cohorts but must also fulfil a minimum effect size cut-off. There is no experimental value in studying a rare variant with an odds ratio of less than 1.5.

Response: This is a good point by the reviewer. We did not mean to imply that the significantly associated genes in this analysis account for the remainder of phenotypic variation observed in the Mean Spherical Equivalent measure of refractive error and have clarified this in the revised manuscript. We agree that effect sizes are very informative and important information in an association study. For the CMC analysis, we have added the frequencies as well as the beta values for each gene. For the VT analysis, we have included the frequencies and directionality (+/-). VT does not output a beta value, because although it is burden test like CMC, VT uses multiple MAF thresholds, calculating a z score from each regression and taking the maximum with p-values determined by permutation. Both Fisher's method and the ACAT method combine p-values without inputting a beta, so beta is not output. Thus, we note that while effect sizes are indeed important as the reviewer points out, we cannot use them as the sole means to prioritize our genes simply because two of our methods do not use them. This is now stated in lines 324-325.

Reviewer Response: I thank the authors for the clarification and additional tables. While I agree with the Authors' contention that effect sizes cannot be the sole means for gene prioritization, it still **must** be taken into account, alongside strength of association, to enable fellow colleagues in the experimental field to decide whether to work on these targets. As such, please also highlight beta values of genes that are considered important in the main text and present the data in the main Figures, and please discuss whether the beta values affect your gene/pathway prioritization framework in the text. Gene/pathway prioritization should take these beta values into account.

Response #2: We thank the reviewer for this comment. First, we reiterate that the type of meta-analysis that was used for this study (Fisher's method) did not output betas as it just combines p-values; further both the ACAT and VT analyses also do not provide betas or an effects metric, again because they are mostly working solely with p-values. This made it impossible to use betas to prioritize genes since none of the meta-analysis results had betas and even in the single sample analyses most of the significantly associated genes did not have betas. Thus, we can only highlight betas for the individual population level analyses using the CMC test. As suggested by the reviewer, we have created a new table for the prioritized genes (Supplementary Table 28) that explicitly shows the betas. We have also created a new Table 1 in the main text that shows, for our top-ranked candidate genes, the p-values for all tests in the meta-analyses and the individual datasets plus the betas for the CMC test in the individual datasets analyzed here. We also discuss this in detail in the main manuscript on lines (382-400, 411-416; 422-425; 432-433; 441-443; 448-449; 455-459). With regards to the pathway analysis, since betas only appeared in one of the three tests used (and not in the meta-analysis) we thought our best prioritization was biological function. However, we feel that we have given the reader enough information to perform their own prioritization based on effect sizes of the CMC tests if they choose. In addition to the new Supplementary Table 28, we note that Supplementary Table 27, which shows the average Spherical Equivalent of Refraction measure for cases vs. controls for each observed variant in the prioritized genes, is more indicative of the effects of the variants in these genes on SER. This table provides the single-variant association p-values and betas so that experimentalists can decide which of these genes and which rare variants within these genes may be worthy of follow-up. There can be wide variability of effect on SER across rare variants within the same gene, which is not surprising. Because different rare variants exist in the

different populations, it is not surprising if the gene-based betas for a specific gene may vary across the populations depending on which rare variants in that gene occurred in the different population samples. For example, variant 1:215972394 in the USH2A gene has a beta of 5.895 in the EACC dataset in Supp Table 27 with mean SER of -6.7 in persons with at least 1 copy of this variant and only -0.445 in persons without this variant allele but this variant does not exist in the IECC dataset. Furthermore, different rare variants in USH2A in the EACC sample have betas ranging from 5.895 to -1.8 and mean case SERs ranging from -6.7 to 1.1. Similarly, in the IECC sample, the 1:215914831 variant in the USH2A gene had a beta of 2.3 and a mean of -2.125 in carriers and 0.2 in non-carriers but does not exist in the EACC sample. Again, the different variants within USH2A varied widely in effect size in the IECC sample. Thus, it is not surprising that the CMC beta's in these two samples were quite close to 0, despite this gene being shown to be significantly associated with SER using other gene-based tests that allow for different variants with different directions of effect on the trait. We have added more discussion of these types of comparisons of the gene-based CMC test betas with the betas for the individual variants and with the average SER in carriers and non-carriers of the individual variants (lines 382-400, 411-416; 422-425; 432-433; 441-443; 448-449; 455-459).

Comment 3: Given the large number of individuals and SER phenotype information available in the CREAM consortium dataset, the authors should also quantify the refractive error in individuals carrying risk gene variants as an orthogonal way to estimate effect size and prioritize risk genes. High priority genes with large effect sizes will be expected to strongly affect SER.

Response: This is an excellent point by the reviewer. We have created a new table (Supplementary Table 27) that shows the average mean SER for both minor allele carriers and non-carriers for all variants in the six prioritized genes (PDCD6IP, MAPT, CHST6, GRHL2, USH2A, P4THTM). This indicates the relative effect size for carriers of each rare variant compared to those with the homozygous major genotype. This is now clearly stated in lines 385-387. In Supplementary Table 27, it can be seen that the mean SERs of many of these rare variants in our prioritized genes are quite different in carriers versus non-carriers of the rare variants in question.

Reviewer Response: I thank the authors for the clarification and the additional table. As with my response to Comment 3, please highlight these values, explicitly presenting the data in the main text and Figures, and discuss whether these values and accompanying beta values are in concordance and whether these values impact the gene/pathway prioritization framework.

Response: We again thank the reviewer for this comment. We have added both the p-values and betas to Supplementary Table 27 and have a paragraph describing the results on lines 384-395 and have compared them with the gene-based CMC betas in several places where they inform the interpretation of the CMC beta values. We have also highlighted that for most of these prioritized variants, the CMC tests were not significant. Again, while the effect sizes do not directly influence the prioritization (simply because we used a different schema; biological function), we believe that we have now sufficiently reported and discussed both gene-based and single-variant effect sizes for the reader to make their own conclusions. We reiterate that the gene-based CMC effect sizes are not useful to prioritize genes that were only significant with the other tests and that these other genes may contain variants with quite large effects on variation of this quantitative trait, but if some variants lead to more negative SER values and some variants lead to more positive SER values than in non-carriers the gene-based CMC betas will be close to zero as is clearly illustrated for *USH2A* but is true for many of the other genes that were significant with non-CMC tests.

Editor comments:

Do please address comments #2 and #3 raised by Reviewer #2. Both comments contain suggestions on effect sizes, strength of the genetic associations, and how they tie up with disease biology. A humble suggestion on data presentation to highlight effect sizes perhaps could be found in Marouli E et al., Nature 2017 Feb 9;542(7640):186-190 by considering Table 1, main text, with Supplementary Table 18. The editorial team earnestly believe that clarifying both points raised by Reviewer #2 would be very helpful to readers of the journal. It is our view that comment #4 need not be addressed.

Thank you for the helpful suggestions. We have modeled our new Supplemental Table 28 and our Supplemental Table 27 on the Supplemental Tables from the cited Marouli et al. Nature 2017 paper. We have also added a new table (Table 1) in the main text similar to Table 1 in the Marouli et al paper. We have also added additional discussion of these results in the discussion section to try to clarify the points about effect sizes (betas) of individual variants vs betas from a gene-based CMC test.

..